# Critical Insight into Pretransitional Behavior and Dielectric Tunability of Relaxor Ceramics

**DOI:** 10.3390/ma16247634

**Published:** 2023-12-13

**Authors:** Sylwester J. Rzoska, Aleksandra Drozd-Rzoska, Weronika Bulejak, Joanna Łoś, Szymon Starzonek, Mikołaj Szafran, Feng Gao

**Affiliations:** 1Institute of High-Pressure Physics Polish Academy of Sciences, ul. Sokołowska 29/37, 01-142 Warsaw, Poland; sylwester.rzoska@unipress.waw.pl (S.J.R.); joalos@unipress.waw.pl (J.Ł.); 2Faculty of Chemistry, Warsaw University of Technology, Noakowskiego 3, 00-664 Warsaw, Poland; mikolaj.szafran@pw.edu.pl; 3Laboratory of Physics, Faculty of Electrical Engineering, University of Ljubljana, Tržaška 25, 1000 Ljubljana, Slovenia; szymon.starzonek@fe.uni-lj.si; 4State Key Laboratory of Solidification Processing, MIIT Key Laboratory of Radiation Detection Materials and Devices, NPU-QMUL Joint Research Institute of Advanced Materials and Structures (JRI-AMAS), School of Materials Science and Engineering, Northwestern Polytechnical University, Xi’an 710072, China; gaofeng@nwpu.edu.cn

**Keywords:** relaxor ceramics, dielectric properties, critical phenomena, glassy dynamics, modeling

## Abstract

This model discussion focuses on links between the unique properties of relaxor ceramics and the basics of Critical Phenomena Physics and Glass Transition Physics. It indicates the significance of uniaxiality for the appearance of mean-field type features near the paraelectric-to-ferroelectric phase transition. Pretransitional fluctuations, that are increasing up to the size of a grain and leading to inter-grain, random, local electric fields are responsible for relaxor ceramics characteristics. Their impact yields the pseudospinodal behavior associated with “weakly discontinuous” local phase transitions. The emerging model redefines the meaning of the Burns temperature and polar nanoregions (PNRs). It offers a coherent explanation of “dielectric constant” changes with the “diffused maximum” near the paraelectric-to-ferroelectric transition, the sensitivity to moderate electric fields (tunability), and the “glassy” dynamics. These considerations are challenged by the experimental results of complex dielectric permittivity studies in a Ba_0.65_Sr0._35_TiO_3_ relaxor ceramic, covering ca. 250 K, from the paraelectric to the “deep” ferroelectric phase. The distortion-sensitive and derivative-based analysis in the paraelectric phase and the surrounding paraelectric-to-ferroelectric transition reveal a preference for the exponential scaling pattern for *ε*(*T*) changes. This may suggest that Griffith-phase behavior is associated with mean-field criticality disturbed by random local impacts. The preference for the universalistic “critical & activated” evolution of the primary relaxation time is shown for dynamics. The discussion is supplemented by a coupled energy loss analysis. The electric field-related tunability studies lead to scaling relationships describing their temperature changes.

## 1. Introduction

Relaxor ceramics remain a cognitive challenge despite seven decades of research [1,2,3,4,5,6,7,8,9,10,11,12,13,14,15,16,17,18,19,20,21,22,23,24,25,26,27,28,29,30,31,32,33,34,35,36,37,38,39,40,41,42,43,44,45,46,47,48,49,50,51,52,53,54]. The significance of innovative applications extends from varactors, signal tunable filters, phase shifters, and frequency-selective surfaces for conformal antennas to possible electrocaloric effect applications [12,14,16,21,27,28,34,35,41,45,46,47,48,49,50,51]. Their unique dielectric properties, sensitivity to the external electric field, and tunability are essential. The increase in research reports since 2020 (24% rise in 2022 and 70% increase, up to approximately 3600 papers in 2023 [52]) shows the significance of relaxor ceramics. Regarding relaxor ceramics applications, the global market is expected to quadruple between 2022 and 2029 to approximately USD 16 billion [52,53].

The unique properties of relaxor ceramics are mainly related to their ‘dielectric constant’ changes near the paraelectric-to-ferroelectric transition. The “homogeneous,” canonic ferroelectric material is crucial as a reference case. In such a system, the temperature dependence of the dielectric constant $\varepsilon \left(T\right)$ is represented by the Curie–Weiss (CW) equation [2,3,4,5,7,9,10,11,12,14,16,19,21,24,25,28,33,34,35,38,41,43]:(1a)$$\varepsilon \left(T\right)=\frac{A_{CW}}{T-T_{C}},$$
(1b)$$\Rightarrow \frac{1}{\varepsilon \left(T\right)}=A_{CW}^{-1}T-A_{CW}^{-1}T_{C},$$
where $A_{CW}=const$ and $T_{C}$ is the Curie–Weiss critical temperature.

For inherently “heterogeneous” relaxor ceramics, instead of the “infinite” singularity $\varepsilon \left(T\to T_{C}\right)\to \infty$ (Equation (1a,b)), a “diffused” temperature maximum of $\varepsilon \left(T\right)$ appears [1,2,3,4,5,6,7,8,9,10,11,12,13,14,15,16,17,18,19,20,21,22,23,24,25,26,27,28,29,30,31,32,33,34,35,36,37,38,39,40,41,42,43,44,45,46,47,48,49,50,51,52,53,54]. The next unique feature is related to strong changes in the dielectric constant when even a moderate external electric field is applied. This is described by the so-called tunability (T%) [9,12,14,15,16,24,25,27,28,34,35,36,41,43]:(2)$$T\left(\%\right)=\frac{\varepsilon \left(E\to 0\right)-\varepsilon \left(E\right)}{\varepsilon \left(E\to 0\right)}\times 100\%.$$

The dynamics of relaxor ceramics exhibit scaling patterns characteristic of glass-forming systems in the previtreous domain. The hallmark of “glassy” dynamics is the super-Arrhenius (SA) temperature evolution of the primary relaxation time, for which the Vogel–Fulcher–Tammann (VFT) dependence is used as the main replacement equation [1,2,3,4,5,6,7,8,9,10,11,12,13,14,15,16,17,18,19,20,21,22,23,24,25,26,27,28,29,30,31,32,33,34,35,36,37,38,39,40,41,42,43,44,45,46,47,48,49,50,51,52]:(3a)$${\tau \left(T\right)=\tau }_{\infty }exp\frac{E_{a}\left(T\right)}{RT},$$
(3b)$$\Rightarrow \tau \left(T\right)=\tau_{\infty }exp\frac{D}{T-T_{0}}{=\tau }_{\infty }exp\frac{D_{T}T_{0}}{T-T_{0}}.$$
Equation (3a) is the canonic SA relation, with the apparent (temperature-dependent) activation energy $E_{a}\left(T\right)$. Equation (3a) simplifies to the basic Arrhenius equation for $E_{a}\left(T\right)=E_{a}=const$, in the given temperature domain. *R* denotes the gas constant.

For the VFT model equation: $E_{a}\left(T\right)=Dt=\left({RD}_{T}T_{0}\right)t^{-1}$, and $t=\left(T-T_{0}\right)/T$ represents the relative distance from the extrapolated singular VFT temperature $T_{0}$. In glass-forming systems, $T_{0}$ is located below the glass temperature $T_{g}$, which by “convention” is linked to $\tau \left(T_{g}\right)=100s$. The amplitude D=const and $D_{T}$ is called the fragility strength [54,55,56].

Broadband dielectric spectroscopy (BDS) is essential for determining the mentioned properties [55]. BDS output results can be represented as the complex dielectric permittivity: $\varepsilon^{*}\left(f,T\right)=\varepsilon^{\prime }\left(f,T\right)-i\varepsilon^{\prime \prime }\left(f,T\right)$. The real part, $\varepsilon^{\prime }\left(f,T\right)$, enables the determination of the canonic dielectric constant. It is associated with the so-called static domain of $\varepsilon^{\prime }\left(f,T=const\right)$ spectrum, where a frequency shift does not significantly change the dielectric permittivity value. For dipolar dielectrics, it occurs within the 1 kHz<f<10 MHz domain. For lower frequencies (LF), below the static domain, a strong increase in $\varepsilon^{\prime }\left(f\right)$ and $\varepsilon^{\prime \prime }\left(f\right)$ occurs, which is commonly linked to the impact of ionic contamination [55]. The response related to relaxation processes appears in higher frequencies, above the static domain [55].

In relaxor systems, for temperature changes in the dielectric constant, $\varepsilon^{\prime }\left(T,f=const\right)$, the diffused maximum near the paraelectric-to-ferroelectric transition is the characteristic feature. Its “branches” are described by the CW relation (Equation (1)) for a set of scanned frequencies. Parameters describing the maximum, $\left(\varepsilon_{max}^{\prime },{\;\varepsilon }_{m}\right)$ and $\left(T_{max},{\;T}_{m}\right)$, are frequency-dependent [1,2,3,4,5,6,7,8,9,10,11,12,13,14,15,16,17,18,19,20,21,22,23,24,25,26,27,28,29,30,31,32,33,34,35,36,37,38,39,40,41,42,43,44,45,46,47,48,49,50,51,52]. This indicates that for relaxors, one should consider the real part of the dielectric permittivity rather than the canonic dielectric constant.

Regarding dynamics, the primary loss curve $\varepsilon^{\prime \prime }\left(f,\;T=const\right)$ characterizing the relaxation process associated with permanent dipole moments is significant. Its time scale estimates the peak frequency, namely the primary relaxation time: $\tau =1/\omega_{peak}=1/2\pi f_{peak}$ [56]. For ferroelectric systems, including relaxors, temperature scans for subsequent frequencies lead to the manifestation of primary loss curves in $\varepsilon^{\prime \prime }\left(f=const,T\right)$ plane. In this case, experimental data are portrayed via the following relation [1,2,3,7,8,9,10,12,13,14,21,24,32,35,47]:(4)$$f\left(T\right)=f\left(T_{m}\right)=f_{\infty }exp\frac{E_{a}^{\prime }\left(T\right)}{T}=f_{\infty }exp\frac{D^{\prime }}{T_{m}-T_{0}},$$
where $T_{m}$ is the temperature for the maximum of $\varepsilon^{\prime \prime }\left(f=const,T\right)$ detected in temperature scans for subsequent frequencies *f*; $T_{m}=T_{max}$ is the temperature describing the loss curve maximum for such scans.

Equation (3a,b) convert into Equation (4) for $\tau \left(T\right)\to 1/f\left(T\right)$ and $T\to T_{m}$.

The “glassy”, “previtreous”-type dynamics is also associated with the non-Debye, multi-time distribution of relaxation times. It is manifested by the “broadening” of the primary loss curves above the single-relaxation-time Debye pattern. Most commonly, it is represented via the Havriliak–Negami (HN) relation [3,4,9,12,13,14,16,17,18,19,21,24,29,30,31,32,33,47,54,55,56]:(5)$$\varepsilon^{*}\left(f\right)=\varepsilon_{\infty }+\frac{\Delta \varepsilon }{{\left(1+{\left(i\omega \tau \right)}^{a}\right)}^{b}}$$
where power exponent 0<a,b<1.

When a,b=1, Equation (5) is simplified to the basic Debye equation associated with a single relaxation time. In Equation (5), $\Delta \varepsilon =\varepsilon -\varepsilon_{\infty }$ is called the dielectric strength and describes the dipolar contribution to the “total” value of the dielectric constant; $\varepsilon_{\infty }$ is the non-dipolar permittivity related to electronic and atomic contributions.

Studies in supercooled glass-forming liquids have shown that the power exponents in Equation (5) can be used as metrics for the distribution of primary relaxation times. It is well illustrated by the Jonsher scaling [57] of primary loss curves $\varepsilon^{\prime \prime }\left(f,\;T=const\right)$ [56,58]:(6a)$$\frac{\varepsilon^{\prime \prime }\left(f<f_{max}\right)}{\varepsilon_{max}^{\prime \prime }}={a\prime f}^{m}\;\Rightarrow \;{log}_{10}\left(\varepsilon^{\prime \prime }\left(f\right)/\varepsilon_{max}^{\prime \prime }\right)={log}_{10}a\prime +m{log}_{10}f,$$
(6b)$$\frac{\varepsilon^{\prime \prime }\left(f>f_{max}\right)}{\varepsilon_{max}^{\prime \prime }}={b\prime f}^{-n}\;\Rightarrow \;{log}_{10}\left(\varepsilon^{\prime \prime }\left(f\right)/\varepsilon_{max}^{\prime \prime }\right)={log}_{10}b\prime -n{log}_{10}f,$$
where T=const, a′,b′=const, and m, n are parameters describing the distribution.

The following link between the distribution metric for HN Equation (5) and Jonsher Equation (6) takes place: m=a and n=ab. The reference Debye relaxation is related to m=n=1.

The analysis based on Equation (6a,b) enables the reliable determining of the relaxation time, using the condition: $d{log}_{10}\left(\varepsilon^{\prime \prime }\left(f\right)\right)/d{log}_{10}f=0$, for $f=f_{peak}$ and $\tau =1/2\pi f_{peak}$. Alternatively, the relaxation time can be determined using the HN Equation (5), but it is associated with the five-parameter nonlinear fitting.

For glass-forming systems, the SA (Equation (3a,b)) and the non-Debye (Equation (5)) behavior take place on cooling from the ultraviscous/ultraslowed domain to the amorphous solid glass. It is associated with the time-scale $\tau \left(T_{g}\right)\sim 100s$ [58]. For relaxor systems, the transition is associated with the “diffused” paraelectric–ferroelectric transition, and the mentioned “glassy” time scale is not reached [1,2,3,4,5,6,7,8,9,10,11,12,13,14,15,16,17,18,19,20,21,22,23,24,25,26,27,28,29,30,31,32,33,34,35,36,37,38,39,40,41,42,43,44,45,46,47,48,49,50,51,52].

It should be stressed that such features of the complex dynamics are absent for basic “homogeneous” ferroelectric systems.

In relaxor ceramics, the temperature at which the distortion from the CW behavior (Equation (1a,b)) occurs on cooling towards the transition is called the Burns temperature ($T_{B}$) [3,4]. It is linked to the onset of polar nanoregions (PNRs), a key concept to explain unique relaxors’ features [3,4,5,6,7,8,9,10,11,12,13,14,15,16,17,18,19,20,21,22,23,24,25,26,27,28,29,30,31,32,33,34,35,36,37,38,39,40,41,42,43,44,45,46,47,48,49,50,51,52,53,54]. It is noted that [37] *“(…) the emergence of PNRs begins to form rapidly through the interaction among adjacent dipoles and orients between the states with the same energy and contributes less to the dielectric permittivity because of violent thermal fluctuation*”. It is suggested that the enhanced interactions between the dipole clusters increase the correlation length, supporting the PNR-associated local field properties. The electric field can reorient the PNRs, which can significantly change the dielectric permittivity behavior, characterized by the aforementioned deviation from the Curie–Weiss law. Following this picture, the “microscopic fluctuations”, in which local fluctuations, associated with the PNRs, cause local changes in the Curie temperature $T_{C}$, often heuristically commented as the consequence of local concentration changes, have been introduced [2,4,5,6,7,8,9,10,11,12,13,14,16,18,20,21,24,25,26,31,32,33,34,35,36,37,38,39,47]. Assuming a Gaussian-type distribution of $T_{C}$, Uchino and Nomura proposed the following relation to describe the dielectric constant changes in relaxor ceramics [2,24]:(7)$$\frac{1}{\varepsilon \left(T\right)}=\frac{1}{\varepsilon_{m}}exp\left(-\frac{{\left(T-T_{A}\right)}^{2}}{2\sigma^{2}}\right)\approx \frac{1}{\varepsilon_{m}}\left[1+\frac{{\left(T-T_{A}\right)}^{2}}{2\sigma^{2}}+\ldots \right],$$
for $T>T_{m}$, $\varepsilon_{m}=\varepsilon_{max}$. Uchino and Nomura proposed the assumption of $T_{A}=T_{m}$ and generalize the above relation to arbitrary power exponent 1≤γ≤2, which led to the commonly used semi-empirical relation [24]:(8)$$\frac{1}{\varepsilon \left(T\right)}-\frac{1}{\varepsilon_{m}}=C^{\prime -1}{\left(T-T_{m}\right)}^{\gamma }\prime .$$

The above relation can be used to represent experimental data in the paraelectric phase, even for $T-T_{m}>1÷3K$ [2,4,5,6,7,8,9,10,11,12,13,14,16,18,20,21,24,25,26,31,32,33,34,35,36,37,38,39,42,43,44]. It should be emphasized that Equation (8) “reduces” to CW Equation (1) for γ=1. It leads to the conclusion that in such a case $\gamma^{\prime }\to 1$ and $T_{m}\to T_{C}$, $\varepsilon_{m}\to \infty$. However, in the opinion of the authors, the link between the exponent $\gamma^{\prime }$ in Equation (8) and well-defined critical exponents [59,60,61,62,63,64] is not clear.

In ref. [7], a different pattern of dielectric constant changes was proposed. The model considered the impact of relaxation polarization processes associated with PNRs by introducing two contributions. The first is due to the thermally activated flips of the polar regions, and the second represents the “other” polarization process. The following relation was obtained for $T>T_{m}$ [7]:(9)$$\varepsilon \left(T\right)=\varepsilon_{\infty }+\varepsilon_{ref.}exp\left(a-bT\right),$$
where in the given case parameters a,b=const, and the coefficient *b* is related to the rate of PNRs in the material.

The behavior in the ferroelectric state, for $T<T_{m}$, was also derived [7]:(10)$$\varepsilon \left(T\right)=\varepsilon_{\infty }+A\left(T\right)\left(ln\omega_{0}-ln\omega \right),$$
where $\omega_{0}$ is the average relaxation frequency of a polar unit cell that is independent of the temperature, i.e., $ln\omega_{0}=const$, *A*(*T*) is an intrinsic parameter of the relaxor material.

Notwithstanding, experimental results for relaxor systems are commonly scaled via Equations (7) or (8) or their parallels.

The reorientation of the PNRs, characterized by the relaxation time (τ), is also the reference for models focused on the non-Arrhenius behavior of the primary relaxation time, for which the VFT relation is used as the scaling reference. For the authors, a problem arises when taking into account that the PNRs concept is coupled to the Burns temperature, i.e., the onset of the distortion from the CW behavior on cooling towards the paraelectric–ferroelectric transition, whereas the glassy dynamics, represented by Equation (3), are observed on both sides of $T_{B}$ [1,2,3,4,5,6,7,8,9,10,11,12,13,14,15,16,17,18,19,20,21,22,23,24,25,26,27,28,29,30,31,32,33,34,35,36,37,38,39,40,41,42,43,44,45,46,47,48,49,50,51,52].

Despite decades of studies, a commonly accepted model to explain the aforementioned characteristics of relaxor ceramics is still lacking [1,2,3,4,5,6,7,8,9,10,11,12,13,14,15,16,17,18,19,20,21,22,23,24,25,26,27,28,29,30,31,32,33,34,35,36,37,38,39,40,41,42,43,44,45,46,47,48,49,50,51,52]. The combination of “glassy” dynamics and the “distorted critical-like” behavior (Equations (1) and (7)) remains a challenge. Even the coherent addressing of the canonical experimental features mentioned above, which are checkpoints for theoretical models, remains a problem. Simple and fundamentally justified scaling dependencies support relaxor ceramics’ modeling, and they can be crucial for the expected boost in innovative relaxor-based devices [52,53].

In this report, we propose looking at the discussed properties of relaxor materials from a slightly different perspective, namely with an explicit reference to the basics of *Critical* Phenomena and Phase Transitions Physics [59,60,61,62] and Glass Transition Physics [54,55,56] and then confront the emerging conclusions with existing and new experimental results, also based on research carried out specifically for this work.

## 2. Materials and Methods

The BST sample was prepared using BaCO_3_ (>98%, Chempur, Piekary Śląskie, Poland), SrCO_3_ (>98%, Chempur, Piekary Śląskie, Poland), and TiO_2_ rutile (>99.9%, Sigma-Aldrich, St. Louis, MO, USA). The materials in stoichiometric proportions (Ba_0.65_Sr_0.35_TiO_3_) were ball-milled for 7 h in water and ethanol, subsequently dried and calcined at 1050 °C for 2 h, and finally, barium strontium titanate was synthesized in a high-temperature solid-state reaction carried out at 1340 °C for 2 h. The sintered material was ground with water and zirconia grinding media on a Witeg BML-6 ball mill at a speed of 300 rpm for 7 h. After drying, the samples of diameter d = 20 mm and height h = 5 mm were obtained via pressing and sintering at 1300 °C for 1 h.

The densities of the samples were measured using a helium pycnometer AccuPyc II 1340 (Micromeritics, Norcross, GA, USA). The density of the synthesized powder was 5.629 ± 0.004 g/cm^3^, while the density of the sintered sample was 5.612 ± 0.005 g/cm^3^.

The average particle size measurements were performed using a Laser Scattering Particle Size Distribution Analyzer LA-950 (HORIBA, Kyoto, Japan). Figure 1 shows the particle size distribution of synthesized barium strontium titanate powder. The average particle size was 1.88 μm.

The powder consists of two fractions: small particles (0.06–0.13 μm) and larger agglomerates (2–7 μm), which were probably formed by the re-aggregation of small particles during the milling process. The powder X-ray diffraction patterns were recorded at room temperature on an X’PERT PRO MPD X-ray diffractometer (Panalytical, Almelo, The Netherlands) with a Cu anode. An X-ray diffractogram was made in the angular 2θ range from 5 to 81° for the powder sample and used to identify the phase composition. It was quantitatively analyzed using the Rietveld method, which was also employed to calculate the size of the crystallites.

A sample holder with a spinner was used in this study. The size of the crystallites and lattice distortions were determined directly from the Sherrer equation for 110 BST reflex. The coarse-crystalline calcite of natural origin and its reflex 104 were used as a half-width standard for the measuring system. The unit cell parameters were refined using the Rietveld method in quantitative analysis. The results of XRD qualitative and quantitative analysis are shown in Table 1. The synthesized BST consisted of 99.3% BST in the assumed stoichiometry (Ba_0.65_Sr_0.35_TiO_3_), including cubic (77.1%) and tetragonal (22.9%) phases, and a small (below 1%) addition of cubic BaTiO_3_.

Based on the microstructural observations of the sintered sample (Figure 2) performed with a scanning electron microscope Prisma E (Thermo Scientific, Waltham, MA, USA), the grains grew approximately five times larger. During the sintering process of the agglomerates, pores and grain boundaries disappear so that we can observe sintered agglomerates with a size in the range of 2–10 μm in the sample structure. The visible defects in the sample were probably caused by the grains being torn out when breaking the sample for observations.

The ceramic samples were then sliced into discs of height h = 1 mm for broadband dielectric spectroscopy (BDS) studies [55]. These were carried out using a Novocontrol Alpha-A spectrometer (NOVOCONTROL Technologies GmbH & Co. KG, Montabaur, Germany), which allows for high-resolution studies up to 5–6 of digits permanent resolution over a broad frequency and temperature ranges. The latter was controlled using a Novocontrol Quattro system. The adjustment of the system elements made by the manufacturer allows for the removal of all parasitic capacitances and the direct registration in the representation of the dielectric permittivity: $\varepsilon^{*}\left(f,T\right)=\varepsilon^{\prime }-i\varepsilon^{\prime \prime }$. The results were recorded isothermally for about 63 different frequencies for 250 temperatures tested successively. It made it possible to analyze the data in the representation $\varepsilon^{*}\left(f,T=const\right)$ as shown in Figure 3, commonly used in Critical Phenomena Physics [59,60,61,62] and Glass Transition Physics [54,55,56,57,58] and in the equivalent representation $\varepsilon^{*}\left(f=cont,T\right)$ (Figure 4) often used in the physics of ferroelectrics [63,64,65,66,67,68] and relaxors [1,2,3,4,5,6,7,8,9,10,11,12,13,14,15,16,17,18,19,20,21,22,23,24,25,26,27,28,29,30,31,32,33,34,35,36,37,38,39,40,41,42,43,44,45,46,47,48,49,50,51,52].

## 3. Results

### 3.1. Model Discussion

#### 3.1.1. “Critical” View on Dielectric-Constant-Related Behavior in Relaxor Systems

The Curie–Weiss (CW) type scaling of dielectric constant temperature evolution is the essential experimental reference for basic “homogeneous” ferroelectrics and related “complex & heterogeneous” ferroelectric relaxor systems [1,2,3,4,5,6,7,8,9,10,11,12,13,14,15,16,17,18,19,20,21,22,23,24,25,26,27,28,29,30,31,32,33,34,35,36,37,38,39,40,41,42,43,44,45,46,47,48,49,50,51,52]. To interpret the CW-type behavior [69,70,71], Devonshire [72,73] directly used the Landau model [74], which considers the free energy power series expansion of the order parameter as the metric of the appearing/disappearing symmetry elements near the continuous phase transitions. Taking the electric polarization P as the order parameter, one obtains [72,73]:(11)$$F=F_{0}+\frac{a}{2}P^{2}+\frac{b}{4}P^{4}+\frac{c}{6}P^{6}-EP,$$
where coefficient $a=A\left(T-T_{C}\right)$; parameters *b* and *c* are considered approximately constant. The last term reflects the interaction with the electric field.

The above relation includes the $\frac{c}{6}P^{6}$ term, characteristic for the tricritical point (TCP) case, the simplest multicritical point associated with meeting three critical points curves [60,61]. The so-called symmetric TCP manifests via the smooth crossover from discontinuous to continuous phase transitions [61]. This term is absent for the basic mean-field (MF) case [60,61]. Equation (11) gives the following pattern for pretransitional changes in the order parameter [59,60,61,62,74,75,76]:(12)$$P\left(T\right)\propto {\left(T_{C}-T\right)}^{\beta }.$$

The exponent β=1/2 for MF and β=1/4 for TCP [60]. For the susceptibility, i.e., the order parameter changes via the coupled external field, χ=dP/dE:(13a)$$\chi \left(T\right)=\frac{a^{-1}}{{\left(T-T_{C}\right)}^{\gamma }},\;\mathrm{for}\;T>T_{C},$$
(13b)$$\chi \left(T\right)=\frac{{\left(2a\right)}^{-1}}{{\left(T_{C}-T\right)}^{\gamma }},\;\mathrm{for}\;T<T_{C}.$$

The susceptibility-related exponent γ=1, both for MF and TCP cases.

Equation (11) leads to the prediction of linear heat capacity changes on both sides of $T_{C}$, i.e., no pretransitional anomaly associated with (critical) exponents but instead only the “jump”: ${\Delta C}_{v}=T_{C}a^{2}/2b$ [61,74]. This behavior does not correlate with experimental results, for which heat capacity pretransitional anomalies have been evidenced [12,14,19]. The basic Landau–Devonshire model dependence (Equation (11)) [72,73], or generally the basic Landau model [74], which originally was exemplified for magnetization and paramagnetic-ferromagnetic transition, is related to the “classical” behavior within the basic MF or TCP approximations, with a hypothetical negligible impact of pretransitional/precritical fluctuations. Nevertheless, such an impact exists. To show it explicitly, Ginzburg supplemented the Landau equation with the gradient term [77,78], which directly recalls fluctuations. Implementing this concept to Equation (11), one obtains:(14)$$F=F_{0}+\frac{a}{2}P^{2}+\frac{b}{4}P^{4}+\frac{c}{6}P^{6}+{\kappa \left(\nabla P\right)}^{2}-EP,$$
where κ is the stiffness coefficient and the term ${\left(\nabla P\right)}^{2}\propto \langle {\delta P}^{2}\rangle$ is related to fluctuations of the order parameter around some “equilibrium” value.

Equation (14) or its parallels for isomorphic critical systems yield temperature characterizations of the correlation length (size) ξ and the lifetime $\tau_{fl.}$ of pretransitional/precritical fluctuations [61]: (15a)$$\xi \left(T\right)=\xi_{0}{\left|T-T_{C}\right|}^{-\nu },$$
(15b)$$\tau_{fl.}\left(T\right)=\tau_{0}{\left|T-T_{C}\right|}^{-\varphi }\propto {\left[\xi \left(T\right)\right]}^{z},$$
where *ν* is the correlation length critical exponent, ϕ=zν; *z* is the so-called dynamic exponent: z=2 for the conserved order parameter and z=3 for the non-conserved order parameter. For the classic behavior (MF, TCP), ν=1/2 and ϕ=1.

Equation (14) leads to the following pretransitional behavior of the heat capacity [61]:(16)$$C_{v}\left(T\to T_{C}\right)\propto {\left|T-T_{C}\right|}^{-\alpha },$$
with exponents α=1/2 ($T<T_{C})$ and α=0 ($T>T_{C})$ for MF; for TCP, α=1/2 both for $T<T_{C}$ and $T>T_{C}$.

Critical exponents are basic parameters characterizing pretransitional behavior. The grand success of Critical Phenomena Physics [59,60,61,62] was related to showing that the values of critical exponents depend only on the space (*d*) and the order parameter (*n*) dimensionalities. Thus, microscopically different systems can be assembled into $\left(d,n\right)$ universality classes, in which isomorphic/equivalent physical properties are described by the same values of critical exponents near critical (singular) points. This universal behavior splits into two categories: (i) non-classical, where the exponents are small irrational numbers, and (ii) classical ones, where the exponents are small integers or their ratios. The latter is associated with space dimensionalities d≥4 (single critical point, MF case), and d≥3 (the simplest multicritical point: TCP) [60,61,62]. The “classical” behavior is also linked to an “infinite” range of intermolecular/inter-element interactions at the microscopic level. One can recall the Ginzburg criterion [77,78] to comment on this issue and the interplay between classical and non-classical criticality. Applying the above discussion to the paraelectric–ferroelectric phase transition, one can relate the classical behavior to the following form of the criterion:(17)$$\frac{\langle {\Delta P}^{2}\rangle }{P^{2}}=\frac{1}{\xi^{d}}\frac{kT\chi }{P^{2}}<1,$$
where P has the meaning of the general order parameter and $\chi \propto {\left|T-T_{C}\right|}^{-\gamma }$ is the order-parameter-coupled susceptibility. 

The Ginzburg criterion [77,78] shows that the classical–non-classical crossover can occur if the space-related range of interactions associated with pretransitional fluctuations becomes smaller than the range of microscopic “permanent” interactions (intermolecular, inter-element) characterizing a given system. This implies that for systems with non-classical critical behavior, the crossover to the classical one should occur far from the critical point, where the correlation length decreases enough. Indeed, such behavior has been evidenced, for instance, a few tens of Kelvin away from the critical consolute temperature in binary critical mixtures of limited miscibility (d=3, n=1 universality class: critical exponents γ≈1.23, β≈0.325, ν≈0.625) [61,76]. However, in critical mixtures, the explicit classical behavior associated with exponents γ=1, β=1/2, ν≈1/2 also has been demonstrated in the broad surroundings of $T_{C}$ under the shear flow [79,80,81,82,83] or under the strong electric field [84,85], with the crossover to the non-classical behavior remote from $T_{C}$. It is a kind of “reversed criticality” under the exogenic uniaxial impact. The mentioned impacts cause the uniaxial elongation of precritical fluctuations, which is possible in the near-critical domain even under moderate external impacts [61]. In the given case, exogenic impacts do not affect intermolecular interactions, and the only factor leading to the “anomalous” appearance of classical behavior may be local uniaxial symmetry, which, in the given case, is induced by exogenic impacts. This concept led to the explanation of changes in the nonlinear dielectric effect (NDE) and the electro-optic Kerr effect (EKE) when approaching the critical consolute point and gas–liquid critical point [84,85]. It also became crucial in explaining the mean-field nature of NDE, EKE, and dielectric constant pretransitional changes in the isotropic liquid phase of nematogenic liquid crystals, where rod-like uniaxial symmetry is the inherent feature [84,85]. Recently, it was also used to show and explain the behavior of NDE, EKE, and the dielectric constant in the liquid phase on approaching the orientationally disordered crystal (ODIC) phase of plastic crystals [58].

For explaining such behavior, it is essential to understand the interrelation between the meaning of the increased dimensionality (d≥4, for MF case) and the “infinite” range of interactions. For both cases, it means that the number of nearest neighbors of a given molecule or element, related to the possibility of interactions (“visibility”), is greater than the number resulting from the simplest “geometrical packing” represented by spheres. This situation can occur when the local symmetry of the elements building the system is predominantly uniaxial. The above allows us to answer a fundamental question:

Why does the wide surrounding of the paraelectric–ferroelectric transition show the mean-field characterizations described by the Curie–Weiss “law” (Equation (1)), related to the MF exponent γ=1?

In our opinion, it can be explained by the inherent uniaxiality, which is the origin of ferroelectricity and is associated with a uniaxial shift in charges within a basic element of the crystalline network.

As for the complex case of relaxor ferroelectric materials, one should take into account their basic material characterization, namely that they are composed of micrometric-size ($l_{Grain}$) grains, connected by “molten” surfaces, that can lead to partially amorphous inter-grain material. Consequently, it can be assumed that in the paraelectric phase of relaxor ceramics on cooling towards the para–ferro transition, first, the “canonical” ferroelectricity develops within grains until the correlation length approaches the grain size. According to the authors, this occurs at the temperature that can be associated with the Burns temperature, then $\xi \left(T_{B}\right)\sim l_{G}$. Further cooling towards the para-ferro transition cannot increase the correlation length of pre-ferroelectric fluctuations up to the infinite value (Equation (15a)), expected for classical, homogeneous ferroelectric systems. However, further cooling towards the transition can improve the pre-ferroelectric ordering within limited grain volumes. Consequently, one can expect the appearance of strong inter-grain local electric fields. They can lead to some coupling of fluctuations confined by grain borders, which can affect their interiors.

At this point, the temperature behavior of the order parameter under the coupled field is worth recalling. For ferromagnetic systems, it is the magnetization and the magnetic field; for ferroelectric systems, it is the electric polarization and the electric field. Under the permanent influence of such a (global or local) field, (electric or magnetic), the order parameter, instead of approaching zero to $T\to T_{C}$ according to Equation (12) shows a strong deviation when passing from the ferro- to the paraelectric phase. It preserves a non-zero value when passing from the high-temperature para- phase to the low-temperature ferro- phase, and vice versa. The onset and the value of this distortion depends on the field intensity. From Equations (11) and (14), the following relations can be obtained for the dielectric constant and the dielectric susceptibility.
(18)$$\chi \left(T,P,E\right)=\varepsilon \left(T,P,E\right)-1={\left(\frac{\partial^{2}F\left(T,P,E\right)}{\partial P^{2}}\right)}^{-1}=\frac{1}{a+3bP^{2}\left(E\right)}=\frac{1}{A\left(T-T_{C}\right)+3bP^{2}\left(E\right)}=\frac{A^{-1}}{T-\left(T_{C}+3A^{-1}bP^{2}\left(E\right)\right)}.$$

The local electric field resulting from the ferroelectric arrangement within grains is not uniform in magnitude and direction. Following Equation (18), one can expect “pseudospinodal singular temperatures” [59,86] corresponding to different available maximum dielectric permittivity values.

It is noteworthy that parallel singular functional forms of pretransitional behavior should appear for Equations (1), (13), (14) and (18) on the approach to the “critical” temperature $T\to T_{C}$ and on the approach to the pseudospinodal temperature, $T\to T_{SP}=T_{C}+\left(3A^{-1}bP^{2}\left(E\right)\right)$ (Equation (18)). However, the latter is associated with finite terminal dielectric permittivity/dielectric constant values. For relaxor ceramics, the “generic” random local electric fields are self-induced on cooling towards the paraelectric–ferroelectric transitions, and the additional “frustration” (F) contribution can also be expected. It can affect the dielectric constant, the dielectric susceptibility, and the singular “critical-like temperature” $T_{SP}=T_{C}+\left(3A^{-1}bP^{2}\left(E\right)+F\right)$.

#### 3.1.2. “Critical” View on Dynamics in Relaxor Systems

“Glassy” dynamics is the next unique feature of relaxor ceramics [1,2,3,4,5,6,7,8,9,10,11,12,13,14,15,16,17,18,19,20,21,22,23,24,25,26,27,28,29,30,31,32,33,34,35,36,37,38,39,40,41,42,43,44,45,46,47,48,49,50,51,52]. It is generally demonstrated by representing the evolution of the primary relaxation time with the VFT relation (Equations (3) and (4)), instead of the simple Arrhenius pattern $\tau \left(T\right)=\tau_{\infty }exp\left(E_{a}/RT\right)$ where $E_{a}=const.$ Non-Debye changes in the shape of loss curves are often described using the HN relation (Equation (5)) [2,6,7,8,12,13,14,15,16,19,21,22,23,24,25,26,31,32,33,34,35,36,37,38,39,47]. Such a scaling pattern is also characteristic of the previtreous domain (i.e., above the glass temperature $T_{g}$) of glass-forming systems. The origin for these universalistic changes, related to $\tau \left(T_{g}\right)<100s$ time-scales, remains a challenge [54,55,56]. For relaxor systems, they are explicitly related to the approach of the paraelectric–ferroelectric transition. The “glassy” dynamics can be associated with the development of the pretransitional fluctuation time-scale (Equation (15b)), which parallels a single-dipole-moment relaxation due to the MF nature of the phenomenon. Below $T_{B}$, which we associate with reaching the grain size limit via the correlation length of fluctuations (Equation (15a)), the increasing influence of the frustration associated with rising impacts of random internal local electric fields may appear. Interestingly, passing through $T_{B}$ the Burns temperature does not affect the parameterization of $\tau \left(T\right)$ using the VFT relation [1,2,3,4,5,6,7,8,9,10,11,12,13,14,15,16,17,18,19,20,21,22,23,24,25,26,27,28,29,30,31,32,33,34,35,36,37,38,39,40,41,42,43,44,45,46,47,48,49,50,51,52].

Recently, it has been shown that the VFT relation is primarily an effective descriptive tool for glass-forming systems, and its fundamental importance is limited [56].

The insight based on the analysis of the apparent activation energy index $I_{L}\left(T\right)=-dlnE_{a}\left(T\right)/dlnT=\left(dE_{a}\left(T\right)/E_{a}\left(T\right)\right)/\left(dT/T\right)$ led to the following expression for configurational entropy changes [56,87]:(19a)$$S_{C}\left(T\right)=S_{0}t^{n}=S_{0}{\left(\frac{T-T_{K}}{T}\right)}^{n}=S_{0}{\left(1-\frac{T_{K}}{T}\right)}^{n},$$
(19b)$${lnS}_{C}\left(T\right)={lnS}_{0}+nlnt\;\Rightarrow \;{\left(\frac{dlnS_{C}\left(T\right)}{d\left(1/T\right)}\right)}^{-1}=\left(\frac{1}{nT_{K}}\right)+n^{-1}T^{-1},$$
where $S_{0}=const$, $T_{K}$ is related to the so-called Kauzmann temperature, the exponent 0.18<n<1.6; the upper limit is related to the dominance of the orientational local ordering (naturally coupled to uniaxiality), and the lower one to the translational order. For n=1, a system has no preferable type of the local symmetry.

This leads to the following “VFT-extended” equation [56,87,88,89]:(20)$$\tau \left(T\right)=\tau_{\infty }exp\left(\frac{D}{T}t^{-n}\right)=\tau_{\infty }exp\left(\frac{DT^{n-1}}{{\left(T-T_{K}\right)}^{n}}\right).$$

This relation correlates with the VFT equation for n=1, but the analysis of experimental data shows that for relaxor systems n>1 [56,87,88,89]. However, the “generalized VFT” Equation (20) contains four fitting parameters, significantly reducing the analysis’s reliability. Obtaining the n parameter independently may be a solution, for example, using the configurational entropy analysis, as defined by Equation (19b). However, it requires high-resolution and long-range experimental results of the heat capacity measurements to determine changes in configurational entropy, which are rarely available.

Recently, a new universalistic description of the so-called steepness index has been shown: $m_{T}\left(T\right)=d{log}_{10}\tau \left(T\right)/d\left(T_{g}/T\right)$. Note that it is proportional to the apparent activation enthalpy $H_{a}\left(T\right)=dln\tau \left(T\right)/d\left(1/T\right)$ [90]:(21)$$m_{T}\left(T\right)=\frac{d{log}_{10}\tau \left(T\right)}{d\left(T_{g}/T\right)}=\frac{1}{T_{g}ln10}\frac{dln\tau \left(T\right)}{d\left(1/T\right)}=C{\times H}_{a}\left(T\right)=C\frac{M}{T-T_{g}^{*}},$$
where C,M=const and $T_{g}^{*}<T_{g}$ is the extrapolated singular glass (vitrification) temperature.

The above relation directly leads to the following three-parameter dependence for the primary relaxation time [90]:(22a)$$\tau \left(T\right)=C_{\Gamma }{\left(t^{-1}exp\left(t\right)\right)}^{\Gamma },$$
(22b)$$ln\tau \left(T\right)={lnC}_{\Gamma }+\Gamma \left(t-lnt\right),$$
where $t=\left(T-T_{g}^{*}\right)/T$ and $C_{\Gamma }=const$.

The number of adjustable parameters can be reduced to only two, since $T_{g}^{*}$ can be easily determined via scaling Equation (21), using the linear regression for the experimental data presented in the plot $H_{a}^{-1}={\left(dln\tau \left(T\right)/d\left(1/T\right)\right)}^{-1}$ vs. T. Knowing $T_{g}^{*}$, one can present the experimental data using the plot defined by Equation (22b), namely $ln\tau \left(T\right)$ vs. t−lnt. We can then use the linear regression fit, and one can estimate the optimal values of $C_{\Gamma }$ and Γ parameters. Thus, the nonlinear fitting can be totally avoided for portraying $\tau \left(T\right)$ changes via Equation (22a).

Equation (22) relates characteristics of the “activated” (i.e., SA-type: Equation (3a)) and the critical-like behavior. It is notable that there is a link between the exponent Γ and the dominant local symmetry in the given system.

If the uniaxial or translational symmetries are dominant, Equation (22a) can be fairly approximated by an even more straightforward critical-like relation [56,90,91]:(23)$$\tau \left(T\right)=\tau_{0}{\left(T-T_{C}^{*}\right)}^{\varphi },$$
where the exponent φ≈9 and $T_{C}^{*}<T_{g}$.

In particular, Equation (23) correlates with the so-called dynamical scaling model (DSM) [92], whose check-point is related to the exponent φ=9. It was suggested to be “universal”, at least for glass forming low-molecular-weight liquids and polymers [92]. Such a statement has not found a reliable experimental confirmation [56]. However, the authors of this work (ADR, SJR) have shown, using a distortions-sensitive analysis, that Equation (23) perfectly describes liquid crystalline (LC) systems, with the inherent uniaxial symmetry of molecules. We emphasize this fact because DSM is the “generic” mean-field model. The classical MF/TCP characterization is also the generic feature of the mentioned rod-like LC systems due to their local uniaxiality [56,90,91].

The discussion presented in this section suggests that the standard VFT relation used to describe “glassy dynamics” in relaxor systems should be considered as an effective tool with limited fundamental significance. The role of the critical-like, mean-field description and the importance of uniaxial symmetry seems to be crucial. It correlates with the discussion of static properties, namely dielectric susceptibility and dielectric constant in Section 3.1.1.

#### 3.1.3. “Critical” View on Clausius–Mossotti Local Field in Ferroelectric Systems

Shortly after Michel Faraday introduced the dielectric constant to characterize the properties of dielectrics, this quantity became important for gaining fundamental insight into the microscopic properties of this type of materials [93]. In 1850, Ottaviano Mossotti proposed the first local field model concept [94]. After supplementations introduced by Rudolf Clausius, it is called the Clausius–Mossotti local field model [95,96]. Further developments of this concept considered a molecule/element inside a cavity in a dielectric, under an external electric field *E* [95,96]:(24)$$F=E+E_{1}+E_{2},$$
where $E_{2}$ is the electric field created by elements/molecules within a semi-microscopic cavity surrounding a given molecule/element, and $E_{1}$ results from charges situated on the surface of the cavity.

For a dielectric system (basically gas or liquid) with a random distribution of elements or a regular crystalline lattice (for solids), $E_{2}=0$. In such a case, summarizing the effect of the cavity surface charge, one obtains [95,96]:(25)$$E_{1}=P/3\varepsilon_{0},$$
where P denotes the polarization vector and $\varepsilon_{0}=8.854\;\left(pFm^{-1}\right)$ is the vacuum electric permittivity.

This approximation can be applied to gaseous dielectrics with non-interacting molecules or non-dipolar liquids [95,96]. Let us recall the dielectric displacement vector: $D=\varepsilon_{0}E+P=\left(\chi^{\prime }+1\right)\varepsilon_{0}E=\varepsilon_{0}\varepsilon^{\prime }E$ and the relation between the polarizability vector and the basic element/molecule polarization:$\;P={\varepsilon_{0}\chi }^{\prime }E=N\alpha_{p}F$, where α means the basic element/molecule polarizability and $N=N_{A}M^{-1}$ stands for the number of base elements/molecules per unit volume, *ρ* is density, M is the molecular mass, and $N_{A}$ is the Avogadro number. Taking this into account, Equation (24) and the fact that $E_{2}=0$, one obtains [95,96]:(26)$$F=\frac{P}{3\varepsilon_{0}}=\frac{\chi^{\prime }}{\chi^{\prime }+3}P\;\Rightarrow \;\;\;\chi^{\prime }=\frac{N\alpha_{P}}{3\varepsilon_{0}}\frac{\chi^{\prime }}{\chi^{\prime }+3}.$$

The re-arrangement of the latter yields:(27)$$\chi^{\prime }=\frac{P}{\varepsilon_{o}E}=\frac{N\alpha_{P}/\varepsilon_{0}}{1-N\alpha_{P}/{3\varepsilon }_{0}}.$$

The above relations (Equations (24)–(27)) show canonic results presented in classic monographs on dielectrics physics [95,96]. Von Hippel [95] supplemented Equation (27) by considering dipolar dielectrics, especially liquid ones, and using the relation introduced by Debye $\alpha_{P}=\mu^{2}/3k_{B}T$. It transformed Equation (27) into the following form [95,96]:(28)$$\chi =\varepsilon -1=\frac{3T_{C}}{1-T_{C}},$$
where $T_{C}=N\mu^{2}/9k_{B}\varepsilon_{0}$.

In his classic monograph [95], von Hippel pointed out the paradoxical consequences of this reasoning for such a common dipolar dielectric liquid as water. He pointed out that it leads to the paraelectric–ferroelectric transition at $T_{C}\approx 1520\;K$, and concluded [95]: “water should solidify by spontaneous polarization at high temperature, making life impossible on this earth!” This paradoxical result is often cited in monographs and presented in undergraduate lectures for students due to its impressiveness and to show the consequences of violating the basic assumptions of a given model. Von Hippel associated the paradox with the lack of short-range interactions associated with the non-zero field *E*_2_. The paradox anomaly for dielectric liquids has been removed by the inclusion of short-range interactions, for example in Kirkwood or Froelich models, commonly used to interpret experimental data for decades [96]. It should be noted that von Hippel’s paradox example ignores an important fact. He considered the density of water for “normal” conditions, i.e., *d* = 1 g/cm^3^ [95,96]. However, for ${T>T}_{C}\approx 1520\;K$ such density is only possible under multi-GPa pressures. It can even lead to exotic properties often detected for materials at extreme pressures.

The following summary from the monograph *Dielectric Physics* by Chełkowski can summarize the considerations regarding the application of the Clausius–Mossotti local field model [96]: “(…) it is obvious that in the case of dipolar materials (…) the Lorentz field model cannot be employed”.

However, there are materials for which the Clausius–Mossotti model can be applied. These are the solid phase of classical ferroelectrics or liquid crystalline ferroelectrics. Both systems are inherently associated with significant permanent dipole moments. Several models deal with this topic [63,64,65,66,67,68] and refs. therein, essentially referring to the qualitative explanation of von Hippel [96], who stated that in ferroelectric materials, an applied electric field or thermal motion can induce a charge displacement and, hence, a net dipole moment within the crystalline network, which can be further increased by the additional displacement caused by inter-ion couplings. The process continues until the thermal motion is overcome at a critical temperature.

But the question remains: Why is the Clausius–Mossotti local field model obeyed in ferroelectric materials? According to the authors, the key argument in favor of such behavior is tautological: it recalls the similarity between the Curie–Weiss and the Mossotti Catastrophe equations, i.e., Equations (1) and (28).

The “Critical” discussion in Section 3.1.1 and Section 3.1.2 can provide an answer. The intrinsic link between the basic mechanism of the appearance of ferroelectricity and the uniaxial symmetry leads to the mean-field characterization. It implies the “immersion” of induced dipole moments in the mean-field surrounding. As a result, a kind of “effective gas” of independent dipole moments can appear, which correlates with the basic assumption of the Clausius–Mossotti local field model. Deviations from this picture can be associated with the emergence of additional specific material properties, e.g., in the broad surroundings of the paraelectric–ferroelectric transition for relaxor ceramics.

### 3.2. Experimental Results and Discussion

The studies were carried out in a Ba_0.65_Sr_0.35_TiO_3_ relaxor ceramic (99.3%). Its preparation and characterization are described in the Methods section. It also includes frequency-related (T=const. Figure 3) and temperature-related (f=const: Figure 4) master plots showing the real and imaginary components of the complex dielectric permittivity: see also Appendix A for the complete data set. The results presented in the Methods section were selected from data covering 250 tested temperatures in the range 123 K<T<373 K, to illustrate general features. The dielectric constant is the basic property whose temperature evolution is considered for relaxor ceramics. However, the canonical definition of the dielectric constant in Dielectrics Physics [95,96] defines it as the nearly constant value of $\varepsilon^{\prime }=\varepsilon$ in the static frequency domain, where a frequency shift has a negligible effect on the measured values. It is visualized as the horizontal domain in $\varepsilon^{\prime }\left(f,T=const\right)$ spectrum. For dipolar dielectrics, it usually occurs for 1 kHz<f<10 MHz [55,58,95,96].

The results presented in Appendix A show that such behavior is almost absent for the tested relaxor ceramics, especially near the paraelectric–ferroelectric transition. The static-type horizontal behavior appears only well above the transition (for the isotherm T=373 K) and for T≈200 K±30. It is noteworthy that the Curie–Weiss temperature $T_{C}\approx 292\;K$.

Consequently, the Curie–Weiss behavior for relaxor ceramics should be discussed in frames of the real part of dielectric permittivity, and the “dielectric constant” should be treated as the replacement name and written “in parentheses”, in the opinion of the authors. For this meaning of “dielectric constant”, the frequency f=10 kHz can be a reasonable choice for the tested system.

Figure 5, Figure 6 and Figure 7 show different aspects of $\varepsilon^{\prime }\left(T,f=10\;kHz\right)$, focused on testing the temperature evolution via a distortions-sensitive and derivative-based analysis (Figure 6) [56,58,90,91,97]. Such an analysis has already been applied in glass-forming systems and “critical” liquids, revealing significant features that are hidden for the direct nonlinear fitting of experimental data [56].

Figure 5 presents the temperature evolution of the “dielectric constant” in the temperature range covering 250 K, including the evolution of its reciprocal. The latter is reminiscent of the commonly used analysis, focused on the Curie–Weiss relation (Equation (1)). It is also used to determine the Burns temperature $T_{B}$, which is related to the distortion from CW behavior when approaching the paraelectric–ferroelectric transition. The departure from CW Equation (1) is gradual, and precise estimation of its value is not possible, namely: $T_{B}=340\;K\pm 5\;K$. Linear changes of $1/\varepsilon \left(T\right)$ in the paraelectric phase can be considered as a confirmation of the process description via the Curie–Weiss Equation (1). It covers the temperature range of about 50 K, although a weak bias appears when approaching the high temperature limit (T≈375 K).

The precise determination of the $T_{B}$ value and the validation of the CW description offers a distortions-sensitive data analysis recalling Equation (1b):(29)$$\frac{d\left(1/\varepsilon \left(T\right)\right)}{dT}=\frac{d\left(A_{CW}^{-1}T-A_{CW}^{-1}T_{C}\right)}{dT}=A_{CW}^{-1}=const.,$$

Such an analysis is shown in Figure 6: the horizontal line, expected according to Equation (29), appears only on the ferroelectric side of the curve related to the paraelectric–ferroelectric transition. There is no horizontal line on the paraelectric side where studies in relaxor systems are focused. Hence, the result validating the Curie–Weiss-type description in the paraelectric phase is negative.

In particular, in the ferroelectric phase, near T≈170 K, a feature of the next phase transition appears. For T>170 K, it follows the pattern parallel to Equation (29), for ca. 40 K.

Figure 7 shows the experimental data from Figure 5 using the semi-log scale. The distortions-sensitive and derivative-based analysis complements the results. It has two aims. The first is to confirm the (surprising) fairly exponential behavior covering the range between T~375 K and T~315 K:(30)$$\varepsilon \left(T\right)=\varepsilon_{ref.}exp\left(a\prime T\right)\;\Rightarrow \;ln\varepsilon \left(T\right)=ln\varepsilon_{ref.}+a\prime T,$$
where $\varepsilon_{ref.},a^{\prime }=const$.

Such behavior is validated by the solid line in Figure 7. It is supplemented by the distortions-sensitive and derivative-based analysis, presented as ${\left[dln\varepsilon \left(T\right)/dT\right]}^{-1}$ vs. T analytic plot. It enables the ”subtle” test of the existence of critical-like domains, described as follows:(31)$$\varepsilon \left(T\right)=\varepsilon^{0}{\left|T-T^{*}\right|}^{-\phi }\;\Rightarrow \;ln\varepsilon \left(T\right)={ln\varepsilon }^{0}-\phi ln\left|T-T^{*}\right|\;\Rightarrow \;\frac{d\left(ln\varepsilon \left(T\right)\right)}{dT}=\frac{-\phi }{\left|T-T^{*}\right|}\;\Rightarrow \;{\left[\frac{d\left(ln\varepsilon \left(T\right)\right)}{dT}\right]}^{-1}=-\phi T\mp \phi T^{*}=a+bT,$$
where $\varepsilon^{0},a,b=const$, $T^{*}$ stands for the critical-like temperature and ϕ is the “critical” exponent.

The above plot also allows the validation of Equation (30), which gives a horizontal line, namely:(32)$${\left[\frac{d\left(ln\varepsilon \left(T\right)\right)}{dT}\right]}^{-1}={\left(a^{\prime }\right)}^{-1}=const..$$

Thus, Equation (30) provides an optimal representation of the experimental data in the paraelectric phase. There is a remarkable agreement between Equation (30) and the output model-relations (Equation (9)), proposed in ref. [7].

Figure 7 also presents the results of the derivative-based analysis of “the dielectric constant” changes in the surrounding of its maximum, associated with the transition from the paraelectric to the ferroelectric phase.

The linear domain detected in such analysis is related to (Figure 8):(33)$$\frac{d\left(ln\varepsilon \left(T\right)\right)}{dT}=a+bT\;\Rightarrow \;d\left(ln\varepsilon \left(T\right)\right)=\left(a+bT\right)dT.$$

The integration of the above yields:(34)$$\varepsilon \left(T\right)=A\exp\left(c+aT+bT^{2}\right)\;\;\mathrm{for}\;285\;K<T<314\;K,$$
i.e., for the surroundings of the paraelectric–ferroelectric transition.

For the paraelectric side of the transition, the following portrayal was validated (Figure 7):(35)$$\varepsilon \left(T\right)=Aexp\left(b+aT\right)\;\;\mathrm{for}\;315\;K<T<375\;K,$$
while for the ferroelectric side of the transition:(36)$$\varepsilon \left(T\right)=\frac{C}{\left|T-T_{C}\right|}\;\;\mathrm{for}\;234\;K<T<285\;K,$$
i.e., correlated with the mean-field Landau–Devonshire model [74,75].

In particular, there are almost no “gaps” between the descriptions related to the following temperature domains.

Temperature changes in the imaginary part of the dielectric permittivity for the discussed “quasi-static” frequency f=10 kHz are shown in Figure 9. This magnitude reflects the energy absorbed for subsequent processes, complementing the message from the scan of the real component, which mainly reflects the appearance and arrangement of permanent dipole moments [58,96]. In the ferroelectric phase, there is a strong manifestation of relaxation processes, which, for $\varepsilon^{\prime }\left(T\right)$ only become explicitly visible only for disturbances-sensitive and derivative-based analysis. This evidence is even stronger, especially in the paraelectric phase, for the dielectric loss tangent $tan\delta =\varepsilon^{\prime \prime }/\varepsilon^{\prime }$, which can be related to the fact that this quantity D=tanδ=(energy lost per cycle/energy stored per cycle).

In Figure 9, the cycle is related to f=10 kHz, i.e., it determines the energy of the process itself, minimizing the influence of the “background”, related to the “whole” system [58,95,96]. This property is also called the dissipation factor, and it is used to define the quality factor Q=1/D, significant for applications in materials engineering.

Figure 9 shows that the tested system is characterized by a relatively low dissipation/loss factor. It increases on approaching the paraelectric–ferroelectric transition, which can be related to an increasing number of permanent dipole moments and ability to interact with the external electric field and is also coupled within multi-element fluctuations. The latter are associated with the anomalously increasing susceptibility χ=ε−1 reflecting the increasing sensitivity of local order parameter changes (polarizability) to the electric field. This effect diminishes away from the transition. The impact on tanδ(T) evolution is shown in Figure 10.

Relaxation times appearing in dielectric permittivity spectra were determined from peak frequencies of loss curves $\tau =1/2{\pi f}_{peak}$, supported by the analysis of $d{log}_{10}\varepsilon^{\prime \prime }\left(T\right)/dT$ and $d{log}_{10}\varepsilon^{\prime \prime }\left(f\right)/d{log}_{10}f$ allowing its unambiguous estimation. This protocol avoids the significant uncertainty for the determination of relaxation times via the Havriliak–Negami relation [58], which requires nonlinear fitting. Such fitting is associated with at least four adjustable parameters, and their number increases to eight when the merging of two relaxation processes creates the loss curve. Loss curves for characteristic temperature ranges, with indications of basic relaxation processes and coupled relaxation times, are shown in Figure 11.

Figure 12 presents the map of relaxation times using the Arrhenius scale: ${log}_{10}\tau \left(T\right)$ vs. 1/T. The inset shows the relaxation time $\tau_{3}$ at low temperatures in the ferroelectric state. It appears that the tested system exhibits a unique feature. Usually, the super-Arrhenius behavior occurs in the paraelectric phase and terminates close to $T_{m}$. For the tested compound it terminates at $T_{term.\;}\approx 330\;K$, considering $\tau_{1}(1/T)$ changes. The super-Arrhenius (SA) behavior of $\tau_{2}(1/T)$ behavior is notable. The SA behavior is related to Equation (22) and has been shown by the apparent activation enthalpy tests, which focused on validating the SA behavior representation by Equation (21). This result is shown in Figure 13. On further cooling towards the transition, a new process emerges. It explicitly follows the simple Arrhenius pattern, with the constant activation energy extending deeply into the ferroelectric state, with no hallmark when passing $T_{m}$ temperature (Figure 12). The height (maximum) of the associated loss curves increases strongly on cooling, as shown in Figure 14. Figure 15 presents the scaled superposition of loss curves related to $\tau_{2}$ relaxation time, showing the essentially non-Debye and broad distribution of relaxation times.

In the ferroelectric phase, phase transformations were detected, as can be seen for temperature evolution of “the dielectric constant” (Figure 7), which suggests a link with the arrangement of permanent dipole moments, and also for $\varepsilon^{\prime \prime }\left(T\right)$ and $tan\delta \left(T\right)$, which may reflect the energy loss associated with these phenomena. The process related to the lowest temperature introduces an additional relaxation time $\tau_{3}$. Its temperature evolution follows the basic Arrhenius pattern, as it is shown in the inset in Figure 12.

For applications of relaxor systems, the sensitivity of the dielectric properties, in particular the ”dielectric constant”, to the external electric field is essential. The fundamental origins of such behavior in relaxor ceramics also have remained a challenge. Figure 16 shows such behavior for the relaxor ceramic discussed in the report. Figure 17 presents the same experimental data, but in respect to the reference at (U=0, E=0): $\Delta \varepsilon \left(E\right)=\varepsilon \left(E=0\right)-\varepsilon \left(E\right)$, i.e., relative changes in “the dielectric constant”. In particular, relatively large changes in “the dielectric constant” occur for relatively weak electric fields in the tested material.

It is also worth noting that a relatively large shift in the $\varepsilon \left(T\right)$ curve maximum reaching $\Delta T\left(E\right)\approx 3\;K$ for the electric field $E=12\;kV{cm}^{-1}$. This shows the notable manifestation of the electrocaloric effect [49] in the given system.

Figure 18 presents the test of the electric field intensity, or the applied voltage, impact on $\Delta \varepsilon \left(E\right)=\varepsilon \left(E=0\right)-\varepsilon \left(E\right)$, in the vicinity of the paraelectric–ferroelectric transition. The red curves show that the following polynomial can describe experimental data:(37)$$\Delta \varepsilon \left(E\right)=\varepsilon_{ref.}+aE^{2}+bE^{4},$$

This report shows that it is possible to describe the temperature changes in the dielectric constant for the following domains: (i) in the ferroelectric phase (Equation (36)); (ii) in the vicinity of the diffused, temperature-stretched, paraelectric–ferroelectric transition (Equation (34)); and (iii) in the paraelectric phase (Equation (35)). The transition to the subsequent domains occurs without a significant temperature gap, which allows us to consider the tunability characterization (Equation (2)), i.e., the relative changes in the “dielectric constant” caused by an external electric field [9,12,14,15,16,24,25,27,28,34,35,36]:(38)$$T=\frac{\varepsilon \left(E\to 0\right)-\varepsilon \left(E\right)}{\varepsilon \left(E\to 0\right)}=1-\frac{\varepsilon \left(E\right)}{\varepsilon \left(E\to 0\right)}.$$

For the ferroelectric side of the para-ferro transition, where the CW Equation (1) is obeyed, one obtains:(39)$$T=1-\frac{A_{CW}\left(E\right)}{A_{CW}}\frac{T-T_{C}}{T-T_{C}\left(E\right)}.$$

It reduces to the temperature-independent parameter $T=1-A_{CW}\left(E\right)/A_{CW}$ if the $T_{C}\left(E\right)$ shift is negligible.

For the paraelectric side of the transition, related to Equation (35), one obtains:(40)$$T=1-\frac{A}{A\left(E\right)}exp\left(\Delta b-\Delta aT\right),$$
where $\Delta a=a\left(E\right)-a$ and $\Delta b=b\left(E\right)-b$, where *a* and *b* are related to E=0.

For the “diffused” surrounding of the para-ferro transition, one obtains:(41)$$T=1-\frac{A}{A\left(E\right)}exp\left(\Delta b-\Delta aT-\Delta cT^{2}\right).$$

### 3.3. Summary of Results

First, a new model for understanding and explaining the *physics of relaxor ceramics* is presented. It is based on references to the fundamentals of *Critical Phenomena Physics* and *Glass Transition Physics*. A particularly notable feature is the focus on the role of local symmetry, in this case uniaxial, which is particularly focused. The proposed model combines the unique properties of ceramic relaxors with the formation of random local electric fields resulting from pretransitional ferroelectric fluctuations inside ceramic “grains”, i.e., confined by grain boundaries. Local electric fields create a specific feedback mechanism that couples the grain-constrained ferroelectric domains and also affect the interior of the grain-constrained ferroelectric domains, causing a bias towards the discontinuous transition. This gives rise to a singular pseudospinodal behavior [89], similar to the critical one, but is associated with an inherent non-zero order parameter (polarization in this case). Another notable feature of the model is the inherent link to basic experimental features. These are characteristic temperature changes in the “dielectric constant”, electric-field-related stability, and glassy dynamics. The latter relates to the natural introduction of non-Arrhenius temperature variations and extreme non-Debye broadening of the primary loss curves.

The model highlights an additional fundamental feature of relaxor ceramics that has been overlooked so far. This is the absence of the “canonic” dielectric constant, defined in *Dielectrics Physics* as the “horizontal” and frequency-independent domain of the real part of the dielectric permittivity spectrum. Therefore, for relaxor ceramics, one should consider the real part of the dielectric permittivity for a selected frequency, and the name “dielectric constant” should be used in quotation marks.

2.The second part of the report presents the experimental results. They are based on relaxor ceramics specially prepared for this work. The innovative differential analysis for the temperature evolution of “dielectric constant” was used. An important result is an explicit demonstration that the “dielectric constant” in the paraelectric phase prefers not the Curie–Weiss portrayal but the exponential description. The results also challenge the “omnipotence” of the VFT relation as the “proof” for the glassy dynamics.3.Based on the results discussed in points (1) and (2), it was possible to describe in a new way the full range of temperature changes of the “dielectric constant” in the para- and ferroelectric phases, for the “diffused” top-domain, and in the ferroelectric phase, virtually without gaps between domains. Equations describing the temperature dependences of the electric-field-related tunability have also been proposed.

## 4. Conclusions

This report presents a model discussion of the unique properties of relaxor ceramics in relation to Critical Phenomena Physics [59,60,61,62], Glass Transition Physics [54,55,56], and the reference to basic “homogeneous” ferroelectrics.

It indicates the importance of pretransitional fluctuations and the importance of uniaxiality in creating mean-field conditions near the paraelectric–ferroelectric transition, in both “homogeneous” and “heterogeneous” (i.e., relaxor ceramics) materials. The discussion includes the extended Devonshire–Landau model [72,73] and some new conclusions for the Clausius–Mossotti [94,95,96] local field model.

It is proposed that random local electric fields between ceramic grains with pre-ferroelectric arrangement, caused by pretransitional fluctuations, are responsible for the generation of characteristic $\varepsilon \left(T\right)$ changes in relaxor ceramics in the broad vicinity of the paraelectric–ferroelectric transition. The action of such local electric fields results in a distribution of local “Curie–Weiss type” domains, associated with a set of pseudospinodal [89] singular temperatures coupled to weakly discontinuous phase transitions:(42)$$\varepsilon \left(T\right)=\frac{A_{Sp}^{local}}{T-T_{Sp}^{local}\left(E\right)}.$$

Pseudospinodal behavior leads to finite $\varepsilon \left(T\right)$ “terminate” values because the discontinuous transition occurs before reaching the singular temperature $T_{Sp}$. It is noteworthy that such a picture allows to avoid problems of the essentially heuristic concept of local critical temperatures $T_{C}$ (Equation (1)) resulting from PNR fluctuations, causing local concentration changes, often recalled in the modeling of relaxor ceramics properties [2,3,4,5,6,7,8,9,10,11,12,13,14,15,16,17,18,19,20,21,22,23,24,25,26,27,28,29,30,31,32,33,34,35,36,37,38,39,40,41,42,43,44,45,46,47,48,49,50,51,52,53,54].

In particular, for basic, “homogeneous”, ferroelectric materials, even a strong external electric field initially leads to non-linear changes in the dielectric constant, which are described by so-called gap-exponents [98]. For relaxor ceramics, a moderate external electric field is already sufficient to strongly decrease the dielectric constant ($\varepsilon^{\prime }$) leading to tunability, which is crucial for applications. The given concept can be associated with the possibility of relatively easy interaction between intergranular electric fields and the external field.

In simple “homogeneous” ferroelectric materials, the static domain manifested via “horizontal changes” in $\varepsilon^{\prime }\left(f,T=const\right)$ scan within the frequency range 1 kHz<f<10 MHz is the common feature. In a static domain $\varepsilon^{\prime }\left(f\right)\approx \varepsilon =const$, despite a frequency shift. This is also the definition of the canonical dielectric constant. For relaxor ceramics, this behavior is absent, and some frequency change of $\varepsilon^{\prime }\left(f\right)$ in the above frequency range is a standard feature. This is shown, for example, in Figure 3 and in Figure A1 in Appendix A. It can also be inferred from numerous reports on relaxor ceramics. In the opinion of the authors, the frequency-dependent “quasi-dielectric constant” is the next hallmark of relaxor ceramics. Such behavior can be deduced from the conceptual model proposed in the report.

For the conceptual model presented, the spatial growth of pretransitional/pre-ferroelectric fluctuations can be estimated by the counterpart of Equation (15a):(43)$$\xi \left(T\right)=\xi_{0}{\left|T-T_{Sp}\left(E\right)\right|}^{-\nu }.$$

This pseudospinodal [60,89] correlation length is limited by the grain size, i.e., $\xi \left(T\right)<l_{grain}$ and additionally influenced by the impact of local electric fields on the singular temperature $T_{Sp}\left(E\right)$. Such size changes are coupled to lifetime changes of fluctuations, which can be expressed by the counterpart of Equation (15b):(44)$$\tau_{fl.}\left(T\right)=\tau_{0}{\left|T-T_{Sp}\left(E\right)\right|}^{-\phi }.$$

Also, in the given case, the terminal values are related to the condition $\xi \left(T\right)\sim l_{grain}$. For the mean-field characterization of the system, the collective and single-element relaxation times are related, i.e., $\tau_{fl.}\propto \tau$. Therefore, the size distribution of grain and the topology, as well as the influence of random local electric fields, must lead to a broad distribution of relaxation times, which is the necessary prerequisite for the glassy dynamics observed in relaxor ceramics, including non-Debye and super-Arrhenius (SA) dynamics.

It is also worth noting that the presented model also explains another characteristic of relaxor ceramics: in different systems, the terminal values of the primary relaxation time range from seconds to milliseconds [1,2,3,4,5,6,7,8,9,10,11,12,13,14,15,16,17,18,19,20,21,22,23,24,25,26,27,28,29,30,31,32,33,34,35,36,37,38,39,40,41,42,43,44,45,46,47,48,49,50,51,52]. For the reference dynamics in glass-forming systems, $\tau \left(T_{g}\right)\approx 100\;s$ [54,55,56].

Experimental studies complemented the model discussion for relaxor ceramics. They were supported by innovative distortions-sensitive and derivative-based data analysis. This was possible due to the specific characteristics of the experiment and the collected data (see Appendix A). Experimental tests were carried out on a Ba_0.65_Sr_0.35_TiO_3_ relaxor ceramic (see Table 1). Figure 5 shows that the permanent increase in the “dielectric constant” takes place from T≈120 K to the maximum reached at $T_{m}\approx 291\;K$, and subsequently $\varepsilon \left(T\right)$ decreases down to $T_{m}\approx 375\;K$. The typical analysis applies a $1/\varepsilon \left(T\right)$ vs. T plot to test the Curie–Weiss (Equation (1)) representation. Such a plot is also shown in Figure 5, suggesting the CW portrayal covers a range from $T_{m}\sim 292\;K$ to T≈228 K, i.e., for ~60 K. in the ferroelectric phase. In the paraelectric phase, which is the particular focus of studies recalling the model analysis, the CW-type (Equation (1)) behavior starts at $T_{B}\sim 235K$ (the Burns temperature) and terminates at T>375 K, i.e., for at least ΔT~40 K. The $\Delta T=T_{B}-T_{m}$ is often considered as one of the metrics of the relaxor-type behavior, indicating the width of the domain that deviates from the CW behavior and is associated with the appearance of polar nanoregions (PNRs), hypothetically responsible for the unique behavior [2,3,4,24]. For the given case, ΔT≈40 K. However, the quality of experimental data enables an effective distortions-sensitive and derivative-based test of the model portrayal, avoiding a parasitic scatter. Figure 6 shows such an analysis focusing on the validation of the mean-field behavior related to the Curie–Weiss Equations (1a) and (13a,b) with the exponent γ=1. The analysis based on Equation (27) explicitly confirms such behavior between T=285 K and T=234 K, i.e., for ~50 K in the ferroelectric phase. In the paraelectric phase, the validation of the CW model description is negative (!).

For the paraelectric phase, the exponential relation Equation (30) gives a much better description of $\varepsilon \left(T\right)$ changes than the CW relation in the temperature range from 375 K to 316 K, i.e., covering ~60 K, as shown in Figure 7. The superiority of the exponential relation (Equation (30)) is demonstrated by the distortions-sensitive analysis (Equations (31) and (32)), with results presented in Figure 8. The exponential relation with the additional temperature term appears on further cooling towards the paraelectric–ferroelectric transition. The obtained scaling patterns in the broad surrounding of the paraelectric–ferroelectric transition are summarized in Table 2.

Other notable results include the “negligible” distance between the domains represented by subsequent scaling relations, the smooth passing of $T_{m}$ when using the (para-ferro) equation and the fact that the crossover from the (para-) to the (para-ferro) domain is associated with the inclusion of a single, temperature-dependent term in the exponential relation.

The question arises as to whether the behavior obtained in the (para-) and the (para-ferro) states might suggest that the “dielectric constant” changes are related to the so-called Griffiths phase [99,100], which is expected for near-critical systems (especially of mean-field type) in the presence of random impacts. In the present case, this is the randomness associated with a random local electric field between grains that can penetrate and affect their interiors. An additional frustration can be caused by changing the properties of intergranular layers.

The dynamics in the paraelectric phase of the tested Ba_0.65_Sr_0.35_TiO_3_ system are somewhat beyond the typical pattern observed in relaxor systems, showing the SA-type behavior commonly represented by the VFT dependence. Such behavior is also observed, but it terminates at ~340 K. The VFT relation can describe it, but the distortions-sensitive analysis prefers the activated-critical description (Equation (20)). At lower temperatures, a new process emerges. The process explicitly shows an Arrhenius-type temperature dependence ($E_{a}=const$), extending from T~330 K to at least T~230 K. Interestingly, this unique pattern for the dynamics seems to have a minimal effect on the “dielectric constant” behavior. The above results are supplemented by $tan\delta \left(T,f\right)$ behavior, focusing on its physical significance and its supporting importance in testing relaxation processes: see Figure 9, Figure 10 and Figure 14, and Appendix A. Finally, the effect of the electric field on the “dielectric constant” was tested, revealing its strong changes for the relatively moderate electric field intensities/voltages. These features are often expressed in terms of tunability (Equation (2)) for practical implementations. The knowledge of the relations describing the broad surroundings of the paraelectric–ferroelectric transitions allowed the derivation of relations for the electric field-related tunability temperature changes (Table 2), without “gaps” between subsequent temperature domains.

In conclusion, we would like to emphasize that the discussion presented in this report has shown the link between relaxor ceramics, Critical Phenomena Physics, and Glass Transition Physics. It shows the importance of uniaxiality for the emergence of mean-field type features. This link suggests that the appearance of a random, strong, intergranular electric field leading to the pseudospinodal behavior [59,86] associated with “weakly discontinuous” phase transitions may be responsible for unique features of relaxor ceramics, particularly regarding the “dielectric constant” ($\varepsilon^{\prime }\left(T\right)$). All of this may re-define the meaning of the Burns temperature and polar nanoregions (PNRs) [2,3,4], suggesting that they are somewhat “effective” and heuristic concepts introduced to explain relaxor systems mystery [2,3,4,5,6,7,8,9,10,11,12,13,14,15,16,17,18,19,20,21,22,23,24,25,26,27,28,29,30,31,32,33,34,35,36,37,38,39,40,41,42,43,44,45,46,47,48,49,50,51,52,53].

The proposed model picture also suggests a significant influence of material engineering characteristics on the dielectric properties of dielectric ceramics. This can be related not only to the grain size, composition, and structure but also to the grain sintering pattern, including the relevant temperature, annealing time, and cooling/heating time rates, which can influence the growth of grains and inter-grain layers, important for the local electric field.

## Figures and Tables

**Figure 1 materials-16-07634-f001:**
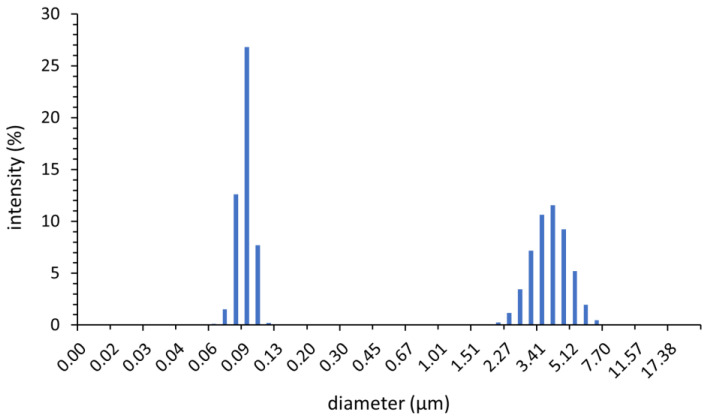
Results of particle size distribution analysis.

**Figure 2 materials-16-07634-f002:**
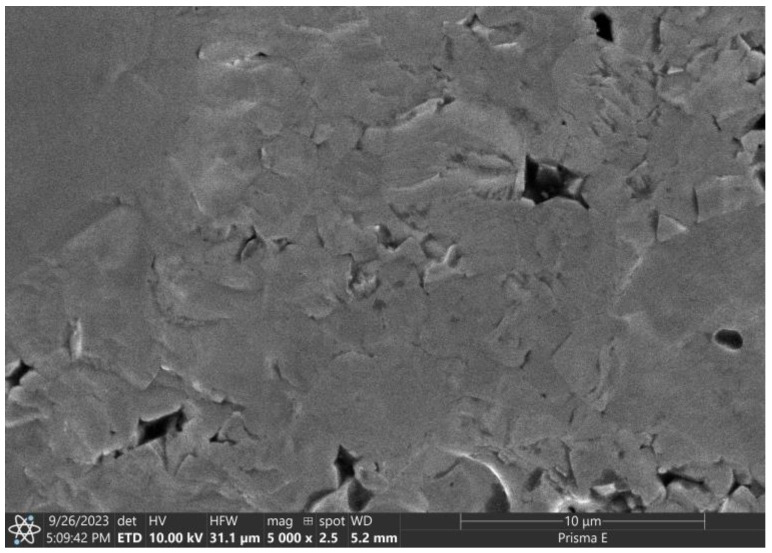
Scanning electron microscope picture of the sintered tested ceramic sample.

**Figure 3 materials-16-07634-f003:**
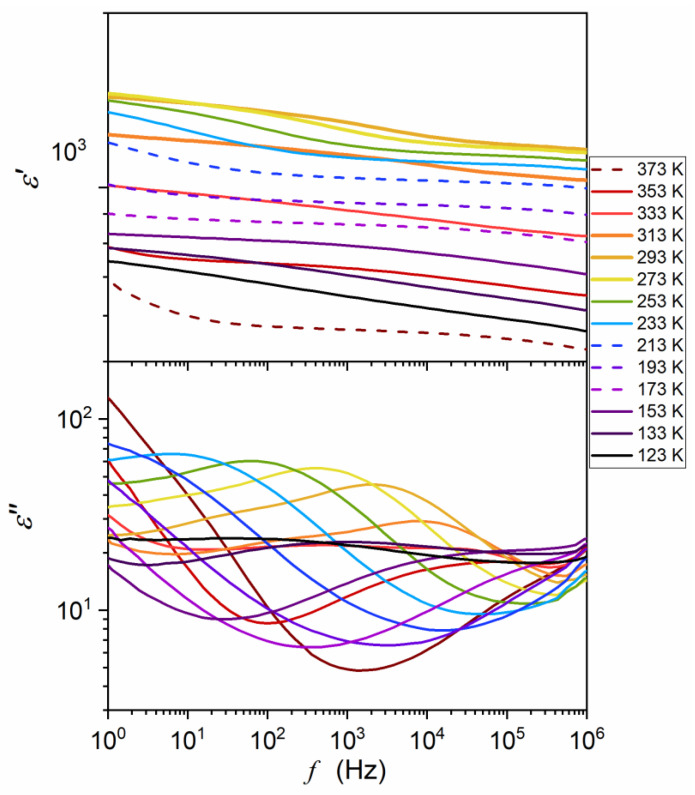
BDS-related spectra showing frequency evolution for the real and imaginary part of the dielectric permittivity (log-log scale), for selected temperatures in the tested ceramic specified in Table 1. The complete data set comprises 250 tested temperatures (see Appendix A).

**Figure 4 materials-16-07634-f004:**
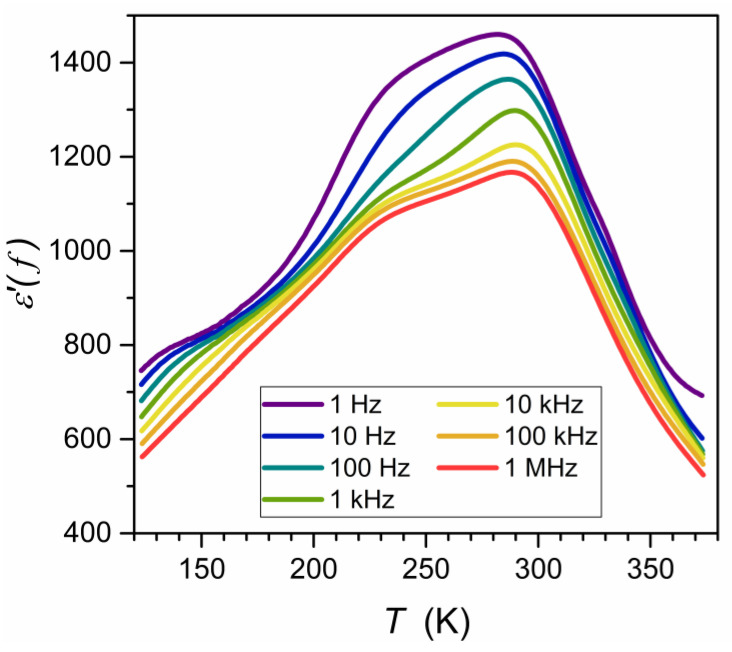
Temperature evolutions of the real part of dielectric permittivity for selected frequencies in the tested relaxor ceramic, specified in Table 1. See also Appendix A.

**Figure 5 materials-16-07634-f005:**
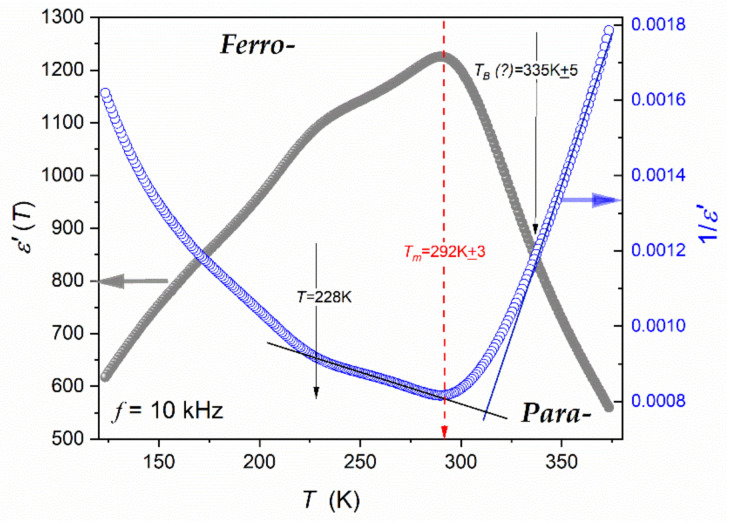
Temperature changes in the real part of dielectric permittivity, related to the so-called “dielectric constant” and its reciprocal. Results for Ba_0.65_Sr_0.35_TiO_3_ relaxor ceramic. The Burns temperature $T_{B}$ is indicated. According to this report, $T_{B}$ is not a significant material characteristic, which is expressed with a “?”.

**Figure 6 materials-16-07634-f006:**
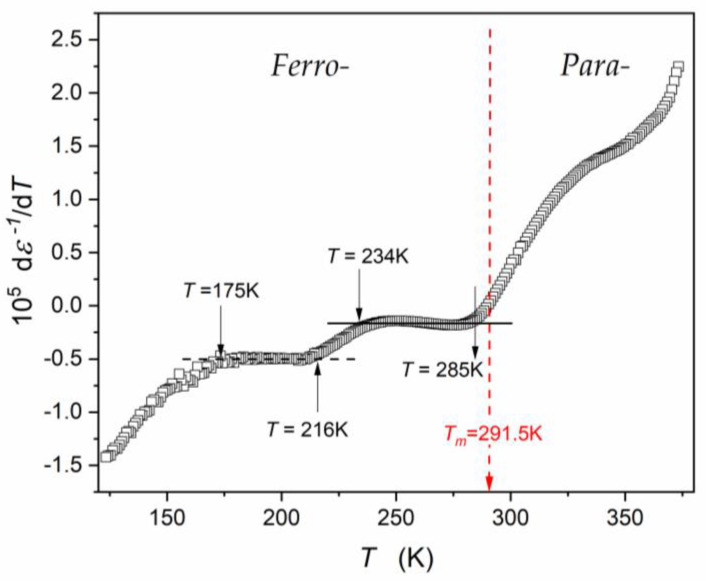
Temperature changes in “the dielectric constant” reciprocal derivative, focused on the distortions-sensitive test of the Curie–Weiss behavior, manifesting via horizontal lines. The analysis for Ba_0.65_Sr_0.35_TiO_3_ relaxor ceramic-based on experimental data is shown in Figure 5.

**Figure 7 materials-16-07634-f007:**
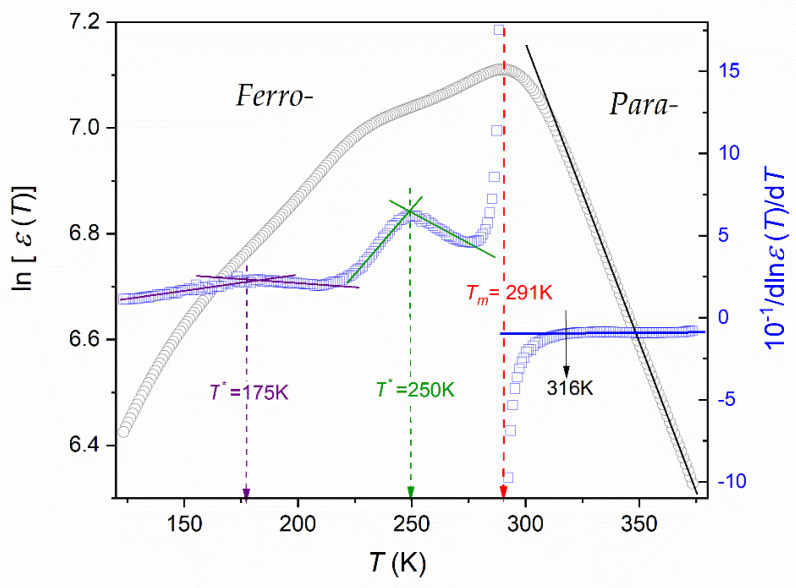
Temperature changes in the logarithm of the “dielectric constant” and the reciprocal of its derivative for the distortions-sensitive test of such behavior, which is manifested by the horizontal line. The analysis for Ba_0.65_Sr_0.35_TiO_3_ relaxor ceramic is based on experimental data from Figure 5.

**Figure 8 materials-16-07634-f008:**
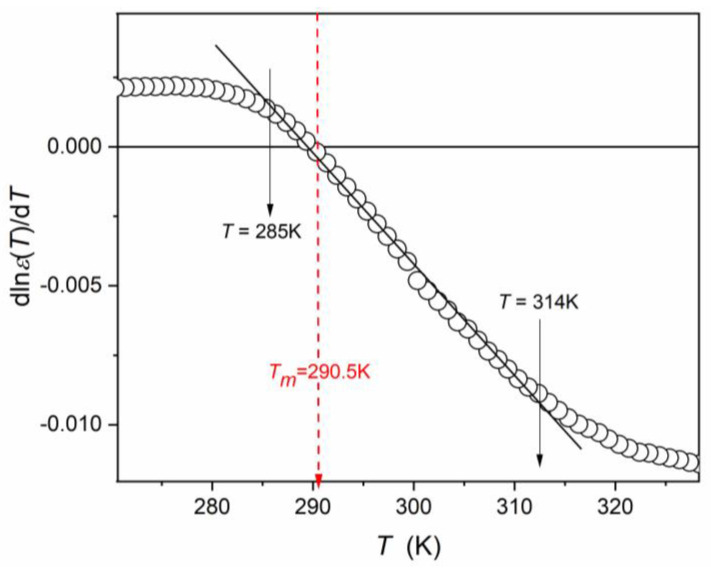
Temperature changes in the derivative of “dielectric constant” ($\varepsilon^{\prime }\left(f=10\;kHz\right)$) logarithm in the surroundings of the paraelectric–ferroelectric transition. The dashed red line indicates the temperature of the “dielectric constant” maximal value. Solid, black arrows indicate terminals of the linear behavior. The analysis for Ba_0.65_Sr_0.35_TiO_3_ relaxor ceramic based on experimental data shown in Figure 5.

**Figure 9 materials-16-07634-f009:**
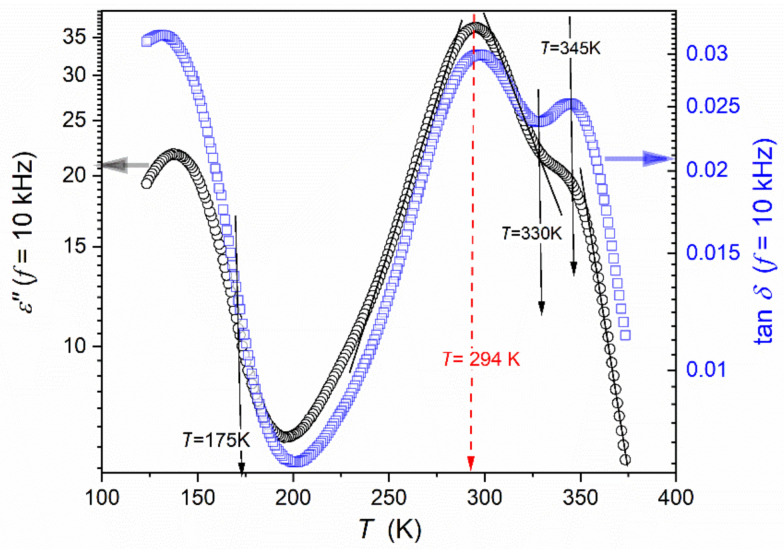
Temperature changes of the imaginary part of dielectric permittivity ($\varepsilon^{\prime \prime }\left(f=10\;kHz\right)$) and related $\tan\varepsilon =\varepsilon^{\prime \prime }/\varepsilon^{\prime }$. The solid, black line indicates characteristic temperatures, and the dashed red line is related to the paraelectric–ferroelectric transition: note a slight shift in comparison with temperatures detected in $\varepsilon^{\prime }\left(T\right)$ analysis. The results are for Ba_0.65_Sr_0.35_TiO_3_ relaxor ceramic (see Figure 3 and Appendix A).

**Figure 10 materials-16-07634-f010:**
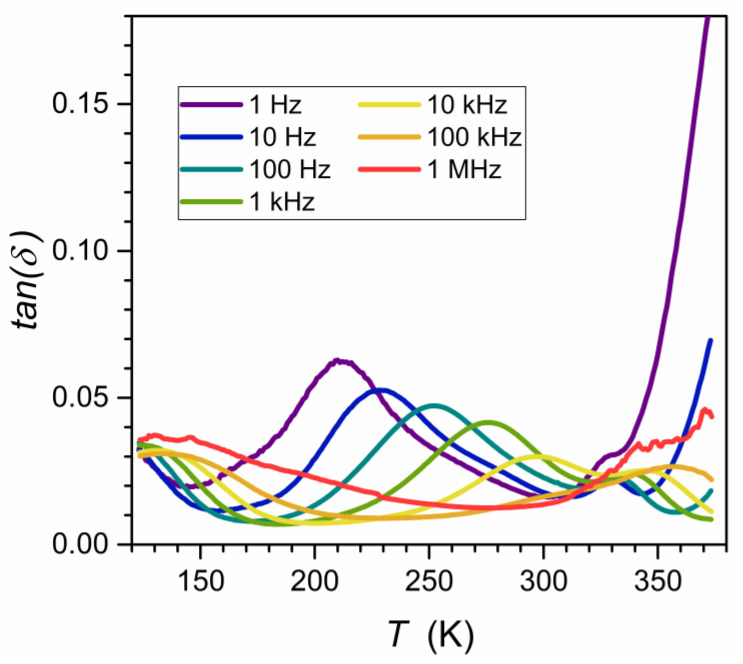
Temperature evolutions of $tg\;\left(T,f=const\right)=\varepsilon^{\prime \prime }\left(f,T\right)/\varepsilon^{\prime }\left(f,T\right)$ for selected frequencies in the tested relaxor ceramic, specified in Table 1.

**Figure 11 materials-16-07634-f011:**
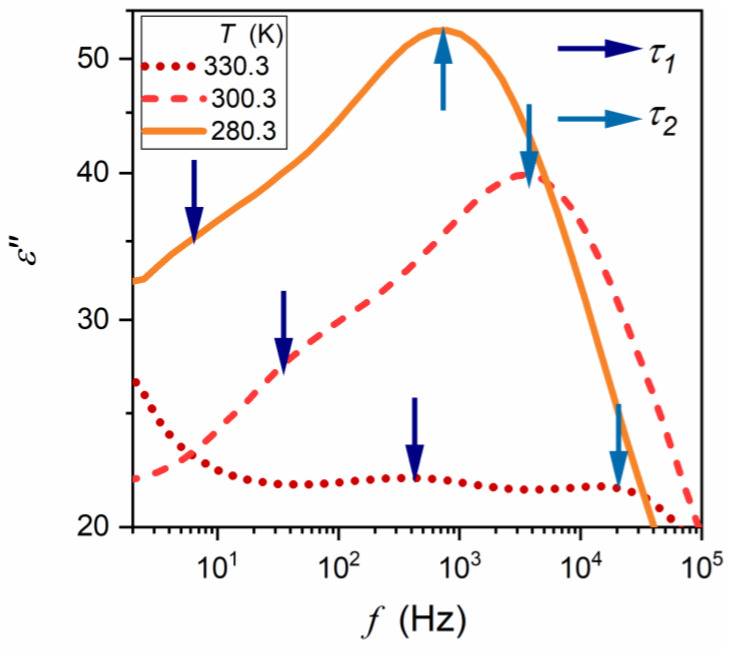
Dielectric loss curves at three selected temperatures in the para- and ferroelectric phases. Relevant relaxation processes are indicated. Results are for Ba_0.65_Sr_0.35_TiO_3_ relaxor ceramic.

**Figure 12 materials-16-07634-f012:**
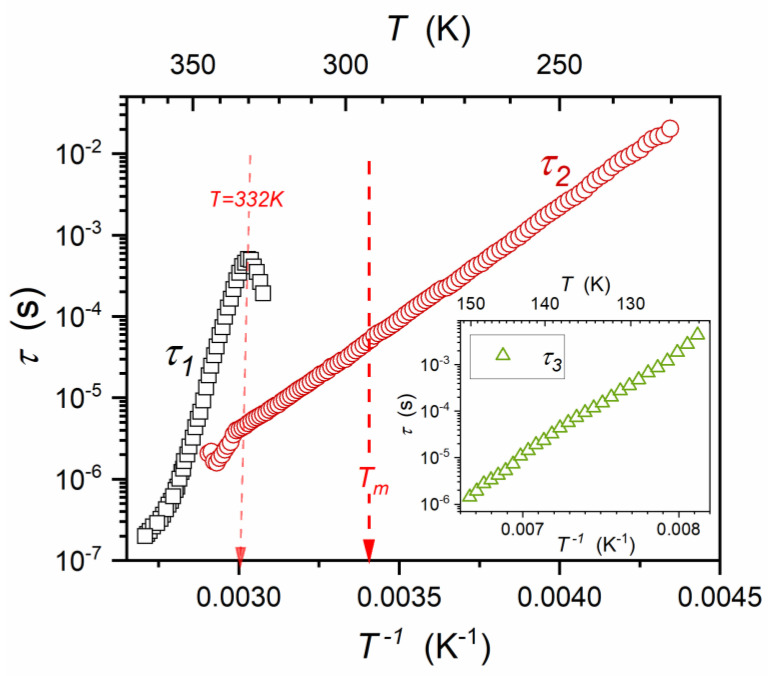
Arrhenius plot of relaxation times detected in Ba_0.65_Sr_0.35_TiO_3_ relaxor ceramics. The inset shows changes in relaxation time of the process emerging in the ferroelectric phase at low temperatures.

**Figure 13 materials-16-07634-f013:**
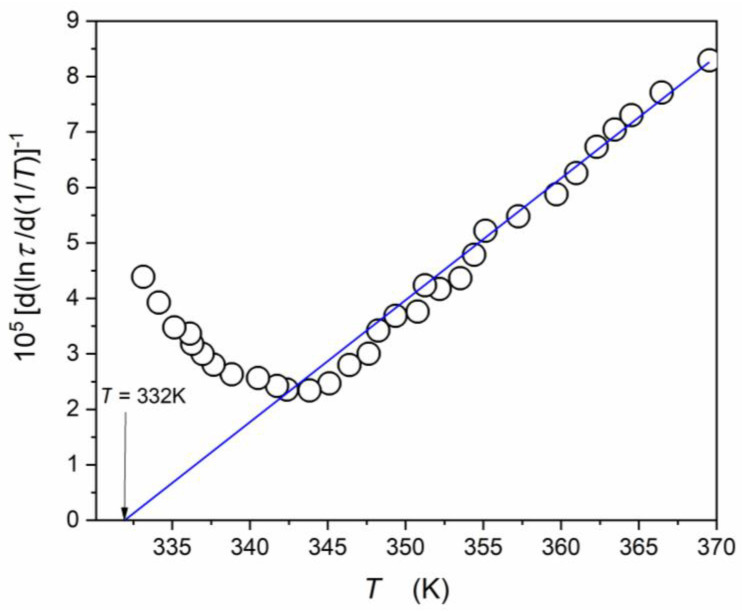
The temperature dependence of the reciprocal of the apparent activation enthalpy focused on validating Equations (21) and (22), which should manifest as linear behavior. The singular temperature $T^{*}$ is indicated by the arrow.

**Figure 14 materials-16-07634-f014:**
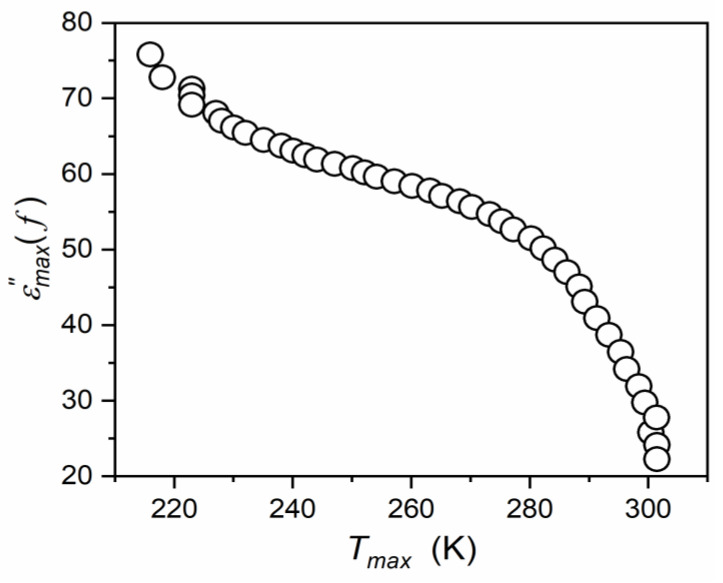
Temperature changes of the maxima of loss curves related to $\varepsilon_{2}$ relaxation time, as indicated in Figure 12.

**Figure 15 materials-16-07634-f015:**
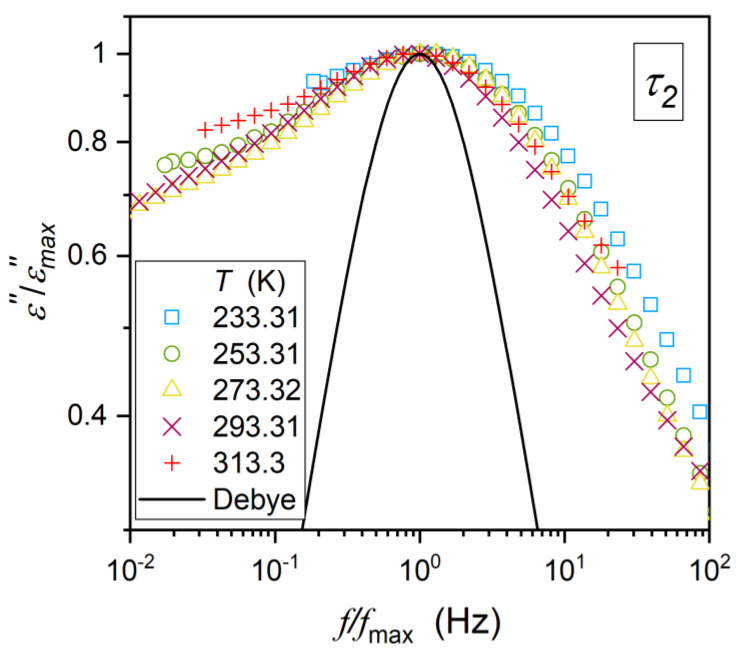
Time–temperature–superposition (TTS) of relaxation time in the tested relaxor ceramic, covering both paraelectric and ferroelectric phases. For comparison, the single relaxation time-related Debye distribution is also shown. The plot is presented in the log-log scale.

**Figure 16 materials-16-07634-f016:**
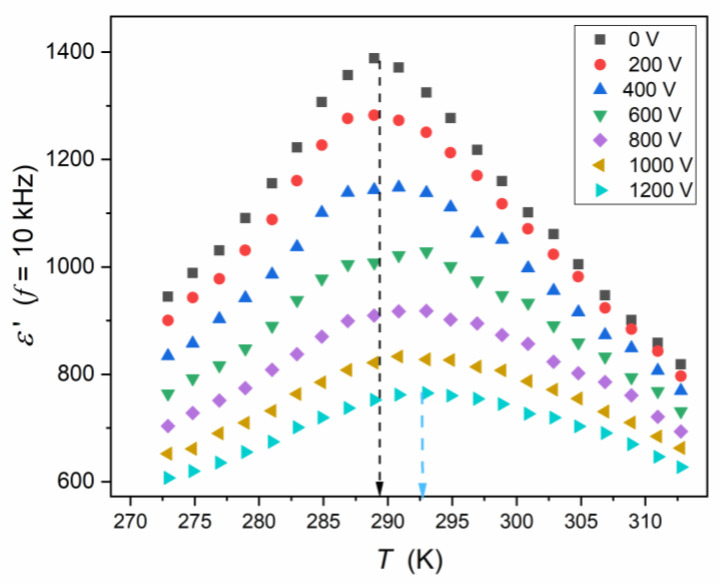
Changes in “dielectric constant” (f=10 kHz) for Ba_0.65_Sr_0.35_TiO_3_ sample, specified in Table 1. The sample was in the form of a disc of height h=1 mm. Measurement voltages are given in the figure. The arrows indicate maximal values.

**Figure 17 materials-16-07634-f017:**
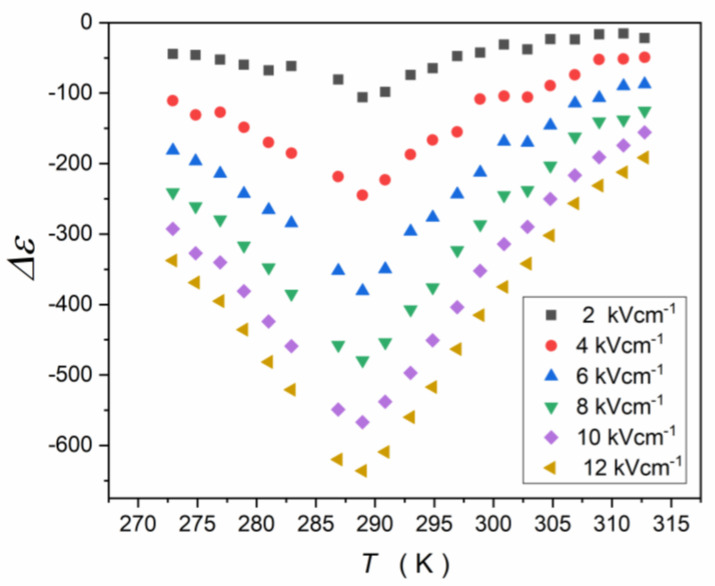
Relative changes in “dielectric constant” (f=10 kHz) for Ba_0.65_Sr_0.35_TiO_3_ sample (specified in Table 1). Scans collected under electric field E≠0 are compared to E=0 behavior.

**Figure 18 materials-16-07634-f018:**
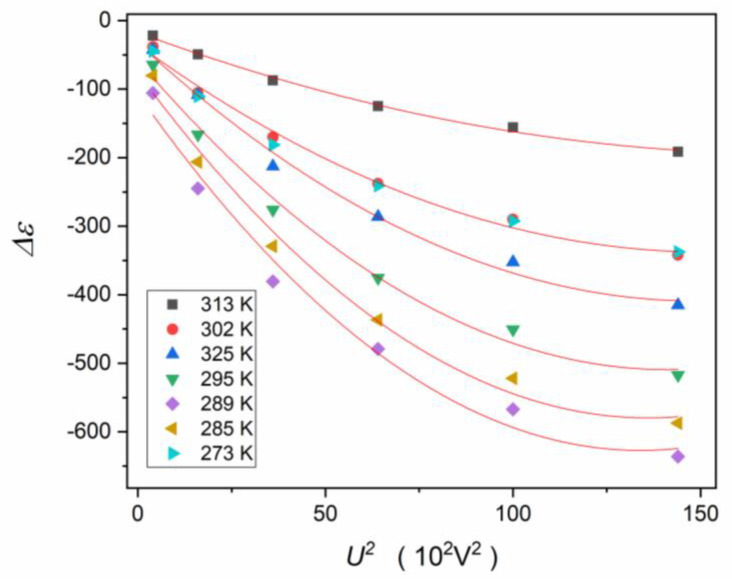
Relative changes in “dielectric constant” (f=10 kHz) in Ba_0.65_Sr_0.35_TiO_3_ sample (specified in Table 1), versus the square of the applied voltage to h=1 mm “thick” sample.

**Table 1 materials-16-07634-t001:** The composition, structure, and size of the tested relaxor ceramic crystallites.

Composition	Share (%)	Crystalline Structure Type	The Share of the Given CS Type %
Ba_0.65_Sr_0.35_TiO_3_	99.3	Cubic	77.1
		Tetragonal	22.9
BaTiO_3_	0.7	Cubic	100

**Table 2 materials-16-07634-t002:** Scaling patterns for temperature changes in “dielectric constant” ($\varepsilon^{\prime }\left(T\right))$ changes in the broad surrounding of the paraelectric–ferroelectric transition in the tested Ba_0.65_Sr_0.35_TiO_3_ relaxor ceramic, specified in Table 1. Note: $T_{m}\approx 292\;K$.

Temperature Range	234 K<T<285 K (ferro-)	285 K<T<314 K (para-ferro)	315 K<T<375 K (para-)
Scaling equation	$\varepsilon \left(T\right)=A_{C}/\left(T-T_{C}\right)$	$\varepsilon \left(T\right)=Aexp\left(c+aT+bT^{2}\right)$	$\varepsilon \left(T\right)=Aexp\left(b+aT\right)$

## Data Availability

Data are contained within the article.

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
