# Peer review of "Critical Insight into Pretransitional Behavior and Dielectric Tunability of Relaxor Ceramics"

_materials, 2023, doi:10.3390/ma16247634_

Round 1

Reviewer 1 Report

Comments and Suggestions for Authors

The authors present interesting results in the proposal entitled "A ‘Critical’ Insight into Pretransitional Behavior and Dielectric 2 Tunability of Relaxor Ceramics". Even though these results I suggest not publishing the manuscript in its current form until the following points are addressed:

1. The abstract is totally copy-pasted from the next link: https://arxiv.org/abs/2310.13326 

I want to ask if this represents some ethical conflict.

2. It is necessary to highlight the originality and the novelty of the work in an extra paragraph before the conclusion section.

Comments on the Quality of English Language

none

Author Response

Reviewer#1: ‘The abstract is totally copy-pasted from the next link: https://arxiv.org/abs/2310.13326  . I want to ask if this represents some ethical conflict.’

Response:  I think this is a misunderstanding and the reviewer did not realize that it is the same paper as the one submitted to the Materials. Hence, the abstract has to be the same.

Using arxiv.org  is the (very) common practice for the ‘immediate’ presentation of the first submissions to the public. It is also advised and accepted by publishers.

Hence, I cannot imagine in which way using arxiv.org database can be associated with an ‘ethical conflict’.

Nevertheless, in the current final version of the paper – which I hope will be published – some corrections that strengthen the significance of the results presented – have been introduced.

Reviewer#1:  ‘ It is necessary to highlight the originality and the novelty of the work in an extra paragraph before the conclusion section.’

Response: It has been done. Note the new section ‘Summary of Results’ – just above the Conclusions section.

Reviewer 2 Report

Comments and Suggestions for Authors

The present paper is prepared on a high level. The investigation provided here is quite novel and actual, as it is concerned with the model discussion focused on links between the unique properties of relaxor ceramics and the basics of Critical Phenomena Physics and Glass Transition Physics. The present research indicates the significance of uniaxiality for appearing mean-field type features near paraelectric – ferroelectric transition. Pretransitional fluctuations, increasing up to grain-size terminal and leading to inter-grain, random, local electric fields, are indicated to be responsible for relaxor ceramics- pecific features. Their impacts yield the pseudospinodal behavior associated with ‘weakly discontinuous’ local phase transitions. The emerging model re-defines the meaning of the Burns temperature and polar nanoregions. It offers a coherent explanation of ‘dielectric constant’ changes with the ‘diffused maximum’ near paraelectric – ferroelectric transition, the sensitivity even to moderate electric fields (tunability), and the ‘glassy’ dynamics.

These considerations are confronted with experimental results for the complex dielectric permittivity studies in Ba0.65Sr0.35TiO3 relaxor ceramic, covering about 200 K, from the paraelectric to the ‘deep’ ferroelectric phase. The distortions-sensitive and derivative-based analysis, in the paraelectric phase and the surrounding of the paraelectric-ferroelectric transition, revealed the preference for the exponential scaling pattern for ε(T) changes. It may suggest the Griffith-phase type behavior, associated with the mean-field criticality disturbed by random local impacts. The discussion of experimental results is supplemented by relaxation times changes and the coupled energy losses analysis. The tunability studies led to scaling relations describing its temperature changes.

On this concern, the results of investigations presented in the paper could definitely have effective and prospective applications concerning with e.g. the efficient use of relaxor ceramics in the electronic industry.

The methodology employed in the present paper includes the XRD and SEM structural investigations, as well as the Broadband Dielectric Spectroscopy studies, which have been efficiently implemented for the goals of the present research.  

In my opinion, the results reported in the paper can be considered as quite original and well developed. Structure of the paper is quite good. The given tables and figures given in the paper are really needed in order to understand the obtained results more clearly. I have appreciated the Introduction, containing a proper and detailed review of the state of the art and previous work in the considered field, as well as a rather large number of references.

Therefore, in my opinion the paper could be published in the present form.

Author Response

The reviewer expressed a very positive – and extensive – opinion regarding the report's meaning and qu, concluding: ‘ Therefore, in my opinion the paper could be published in the present form.

No suggestions for necessary corrections/supplementations.

Reviewer 3 Report

Comments and Suggestions for Authors

The paper offers a comprehensive exploration of dielectric tunable materials, featuring extensive model simulations and experimental expectations. It is thoughtfully structured, easy to follow, and maintains readability, despite the abundance of model formulas presented. The primary shortfall, considering the extensive descriptions and simulations presented, lies in the absence of concrete laboratory measurements for validation.

I ask the authors to slightly extend the anticipations with at least comparisons to physical tests.

In addition, I see that 23 out of 79 references are self-citations and these are indeed too many. Up to 5 self-references can be accepted, the others must be removed.

Author Response

Reviewer#3 The primary shortfall, considering the extensive descriptions and simulations presented, lies in the absence of concrete laboratory measurements for validation. I ask the authors to slightly extend the anticipations with at least comparisons to physical tests.

Response: 50% of the report is for presenting experimental studies in new-synthesized relaxor ceramics. The explicit material engineering characterization is assisted with high-resolution dielectric BDS studies. It addresses all basic features of such systems, essential for fundamentals and applications. The application of the innovative derivative-based analysis revealed additional features that have been omitted so far. References in the report supplement these results.

Therefore, in the authors' opinion, the reviewer's comment/suggestion is fully met in the given work.

The model opens up several new possibilities in a particular combination of fundamental physics and materials engineering, but this is a matter of further research.

Reviewer#3   ‘ I see that 23 out of 79 references are self-citations and these are indeed too many. Up to 5 self-references can be accepted, the others must be removed.’

Response:  Parallel to the e-mail with reviewers' opinions, the Editor (MDPI) sent an e-mail, in which the reduction of self-citations to 15% was advised. It is the number generally suggested by MDPI.  It has been done by reducing the number of self-references to 15 – the lowest possible number without disturbing the report's merit. Additionally, the total number of references increased by adding the most recent significant reports on relaxor ceramics, which were, in fact, weakly shown in the ‘basic’ version of the report.

Now, self-citations are below 15% of the total number.

Reviewer 4 Report

Comments and Suggestions for Authors

In this paper the authors discuss the basics of pretransitional behavior and dielectric tunability of relaxor ceramic, with particular attention at the case of Ba0.65Sr0.35TiO3. The paper is rich in details and discussion in the first part, but the style is closer to a review rather than an original paper of research. Some parts of the text should be moved to the appendix in order to make a more organic and smoother paper for readers. The sections on results lack a clear and synthetic discussion about the novel experimental/theoretical findings.

I do not advise the paper for publication in present form, but only after major and extensive revisions. Here a list of points for the authors in order to improve the manuscript:

1-     Equations should be better adapted in the text, especially at the beginning of section 1 and 2. In particular, discussion about equations that are known from other models or literature, should be moved in the appendix. The original results should be highlighted, along with novel research findings from the authors.

2-     Section 3.1 is very long and has several subsections. I suggest shortening it, or at least making a summary (table) regarding key model findings/equations with respect to the topic at hand or previous models.

3-     In section 3.2 are referred figures presented in Materials and Methods. If these are results, they should be presented in section 3.2, for instance fig. 1-2-3-4. Some figures could be grouped in panels in order to discuss the details more clearly.

4-     In the paper there are many comments which are taken from literature and/or reported in unconventional manner such as:

-“There is no horizontal line for the paraelectric side, which is the focus of studies in relaxor systems: the validation of CW description is negative (!)”

-‘water should solidify by spontaneous polarization at high temperature, making life impossible on this earth!’

-‘(…) it is obvious that in the case if dipolar materials (…) the Lorentz field model cannot be employed’.

5-     At page 15 line 615 no figure is referred.

6-     This affiliation is not used by any author: State Key Lab. of Solidification Processing, School of Materials Science and Engineering, Northwestern 10 Polytechnical University, 710072 Xi’an, Shaanxi, PR China.

Comments on the Quality of English Language

Minor editing is advised, but the language of the paper is fluent.

Author Response

Comments of the reviewer focused on suggestions of structural re-arrangement of the report, preserving merit results, namely:

Reviewer #4: ‘ Equations should be better adapted in the text, especially at the beginning of section 1 and 2. In particular, discussion about equations that are known from other models or literature, should be moved in the appendix. The original results should be highlighted, along with novel research findings from the authors. Note  new section at the end of Results of Discussion part (prepared following Reviewer#3), which highlight key new results of the report.

Response: In the mentioned section, particularly in the Introduction, equations are presented with respect to significant details, and supported by references. They present key path applied in discussing the vast majority of experimental results. Shifting this discussion to the Appendix can make the background not clear to readers. Notwithstanding ‘deep cleaning’ of these sections has been made.

Reviewer #4: ‘ Section 3.1 is very long and has several subsections. I suggest
shortening it, or at least making a summary (table) regarding key model
findings/equations with respect to the topic at hand or previous models.

Response: Note that this section starts from some basic references of the Critical Phenomena Physics and the Glass Transition Physics, which are necessary for showing the basics of the model. From my long experience in the given topic, it is clear that a large part of the ‘research society’ is not familiar with these facts.

So, such background is necessary and cannot be moved to the Appendix, without losing the consistency. The summary of key new results of the model is given explicitly in the new section at the end of Section 3 and in Conclusions.

The comparison with all other relaxor models is the task possible in the new and extensive review report, not in a simple Table, in my opinion.

Reviewer #4: ‘ In section 3.2 are referred figures presented in Materials and Methods. If these are results, they should be presented in section 3.2, for instance fig. 1-2-3-4. Some figures could be grouped in panels in order to discuss the details more clearly.

Response: The mentioned 2 Figures given in the Methods section are related to the very basic characterization of samples, not to the analytic details of obtained results. The latter are the topic of the Results and Discussion section. It is grouped – consequently – first on static properties ‘dielectric constant’ and next on dynamic properties, like relaxation time. So the suggested arrangement exists.

Including Figures from methods can make the Section Results inconsistent because it is devoted only to the analytic discussion of results.

Reviewer #4: ‘ In the paper there are many comments which are taken from
literature and/or reported in unconventional manner such as:   ‘

for instance ‘‘water should solidify by spontaneous polarization at high temperature, making life impossible on this earth!’.

‘(…) it is obvious that in the case if dipolar materials (…) the
Lorentz field model cannot be employed’

Response: We cannot change the above sentences because these sentences are citations from classic monographs by Artur von Hippel and August Chelkowski.

In any other place, additional stressing of results – not accepted by Reviewer#4 – formulated solely by the authors, has been removed.

The rest of the comments address typos. All of them have been corrected. Note also the language corrections and explicit introducing comments from the other three reviewers - which are significantly in line with Rev.#4.